

# Design, Construction and Commissioning of the Braunschweig Icing Wind Tunnel

Stephan E. Bansmer[1], Arne Baumert[1], Stephan Sattler[1], Inken Knop[1], Delphine Leroy[2],
Alfons Schwarzenboeck[2], Tina Jurkat-Witschas[3], Christiane Voigt[3,4], Hugo Pervier[5], and
Biagio Esposito[6]

[1]Institute of Fluid Mechanics, Technische Universität Braunschweig, Hermann-Blenk-Str. 37, Braunschweig, Germany
[2]Laboratoire de Météorologie Physique (LaMP), 63178 Aubiere Cedex, France
[3]German Aerospace Center, Institute of Atmospheric Physics (IPA), Oberpfaffenhofen, 82234 Wessling, Germany
[4]Johannes Gutenberg-University Mainz, Institute of Atmospheric Physics, 55099 Mainz, Germany
[5]School of Aerospace, Transport and Manufacturing, Cranfield University, Cranfield, Beds MK43 0AL, United Kingdom
[6]Italian Aerospace Research Center (CIRA), Via Maiorise, 81043 Capua, Italy

*Correspondence to:* Stephan E. Bansmer (s.bansmer@tu-braunschweig.de)

**Abstract.** Beyond its physical importance in both fundamental and climate research, atmospheric icing is considered as a severe operational condition in many engineering applications like aviation, electrical power transmission and wind-energy production. To reproduce such icing conditions in a laboratory environment, icing wind tunnels are frequently used. Creating and maintaining a stable icing cloud in the tunnel test section yields different design constraints compared to conventional

wind tunnels. In this paper, a comprehensive overview on the design, construction and commissioning of the Braunschweig icing wind tunnel is given. The tunnel features a test section of 0.5m × 0.5m with peak velocities of up to 40m/s. The static temperature ranges from -25°C to +30°C. Supercooled droplet icing with liquid water contents up to 3 g/m$^3$ can be reproduced. Outstanding ability of the tunnel is to simulate ice crystal icing with natural ice crystals for ice water contents up to 20 g/m$^3$. We further show, how current state-of-the-art measurement techniques for particle sizing perform on ice crystals. The

data is compared to those of in-flight measurements in mesoscale convective cloud systems in tropical regions. Finally, some applications of the icing wind tunnel are mentioned.

## 1   Introduction

Atmospheric icing affects the operational performance of many man-made devices and ground structures. Examples include

icing of electrical power networks (Farzaneh (2008)), wind turbines (Hochart et al. (2008)), communication towers (Mulherin (1998)) and aircraft (Gent et al. (2000)).

Based upon the understanding of the physics and chemistry of clouds (Lamb and Verlinde (2011)), several icing mechanisms





are well-known to atmospheric scientists and engineers: supercooled droplet freezing (Jung et al. (2012)) – including freezing drizzle and rain (Zerr (1997)), snow accretion (Makkonen (1989)) and mixed-phase icing (Currie et al. (2012)). Particularly, in the field of aircraft icing, supercooled large droplet (SLD) icing with median droplet diameters above $50\mu$m is considered as an independent category (Politovich (1989)).

Supercooled means that the droplets are in a metastable liquid state although their temperature is below the freezing point. Solidification can take place through homogeneous or heterogeneous nucleation. In absence of any impurities like aerosols, *homogeneous nucleation* starts with exceeding a barrier of free enthalpy triggering the creation of critical embryos. The change of free enthalpy is a function of latent heat of fusion (promoting phase change) and energy needed to create a new water-solid interface (hindering phase change). With increasing supercooling the latent heat of fusion increases thus making homogeneous nucleation more probable. Droplet impact on a solid substrate promotes *heterogeneous nucleation* initiating a fast development of ice dendrites in the liquid (Schremb and Tropea (2016)).

The following important parameters are of particular significance in defining the boundary conditions of the icing process:

– Classical, fluid-mechanical testing parameters like **Reynolds number** $\mathrm{Re} = \frac{\varrho}{\mu} U_\infty c$, **Mach number** $\mathrm{Ma} = \frac{U_\infty}{a_\infty}$ and **angle of attack** $\alpha$, where $\varrho$ is the density and $\mu$ the dynamic viscosity of air, $U_\infty$ the free-stream velocity, $c$ a reference length of the test article and $a_\infty$ the speed of sound.

– Water concentration in the air. On the one hand, this is the **Liquid Water Content** (LWC), a measure of the mass of water per unit volume of air. Typical values for atmospheric icing conditions range from 0.1 g/m$^3$ to 3 g/m$^3$ and have been measured in the updraft of continental convective systems e.g. above the Amazon basin, see Wendisch et al. (2016) and Braga et al. (2017). On the other hand, in the presence of ice crystals, an **Ice Water Content** (IWC) has to be specified. Inside continental and oceanic mesoscale convective systems, IWC-values are about 1 g/m$^3$ (Gayet et al. (2014)). While ice water content in mid-latitude cirrus is generally low (Voigt et al. (2017)), peek values of 6 g/m$^3$ have been observed in deep convective systems. Commonly, a **Total Water Content** (TWC) is defined as the sum of LWC and IWC.

– **Median volume diameter** (MVD) of the statistical distribution of water droplets in the liquid cloud. MVD attempts to reduce the size distribution to a single, representative scalar diameter. In the presence of ice crystals, a **median mass diameter** (MMD) is defined because of their variable density.

– **Static air temperature**. For water droplet icing, static temperatures close to 0°C promote glaze ice formations, whereas rime ice formation becomes predominant at lower static temperatures (Wagner (1997)).

– **Humidity**, i.e. the amount of water vapor in the air, is an important parameter for mixed-phase ice accretion (Currie et al. (2012)). Jung et al. (2012) showed its relevance for pure droplet icing. A distinction is made between absolute humidity AH and relative humidity RH. Absolute humidity is defined by the mass of water vapor per unit volume of air.



With the partial pressure of water vapor $p_{\text{vapor}}$, the static temperature $T$ of air and the specific gas constant of the vapor $R_{\text{vapor}}$, this yields $\text{AH} = \frac{p_{\text{vapor}}}{R_{\text{vapor}} T}$. Relative humidity is defined as the ratio of the partial pressure of water vapor $p_{\text{vapor}}$ to the equilibrium vapor pressure of water $p_{\text{vapor}}^*$, i.e. $\text{RH} = \frac{p_{\text{vapor}}}{p_{\text{vapor}}^*}$.

– **Accumulation time** $t_{\text{acc}}$, for which the model is exposed to the cloud of super-cooled droplets and/or ice crystals.

Among the above enumeration of boundary conditions, temperature is of particular importance, because it governs the freezing dynamics of impacting super-cooled water droplets. At very low temperatures, the droplets will solidify shortly after their impingement and entrap the surrounding air. The resulting ice accretion takes is called rime ice, see figure 1. With increasing temperature, the solidification process of impacting droplets is retarded, yielding wind-driven water film dynamics. At locations with increased convective heat transfer, the water film freezes, yielding a nearly transparent – glaze ice shape with typical
horn formations.

The idea of using a MVD as a representative scalar for approximating the collision efficiency of an entire droplet spectrum is attributed to LANGMUIR (Langmuir (1961); Finstad et al. (1988)). His theory of droplet trajectories relates droplet size distributions and ice accretion on a cylinder. Having mounted such a cylinder on an airplane that flew through a cloud of super-
15 cooled fog, Langmuir derived an assumption on the droplet size spectra, which can be found in table 1. The term 'median' results from the median value of the liquid water content in this distribution, where $d/\text{MVD} = 1$. Langmuir himself described the MVD as a diameter "such that half of the liquid water content of the fog is in droplets of larger [diameter] and the other half in droplets of smaller [diameter]."

For measuring the droplet size spectra, a broad field of instrumentation is available (Tropea (2011)), with particular developments to study cloud physics (Cober et al. (2001)). Measurement principles are oftentimes based on light scattering and include direct imaging, interferometry, time shift and pulse delay, see also chapter 4. Usually, these measurement techniques are evaluating the size of individual droplets in a small measurement volume over time. Based on a sufficiently large sample of droplets ($> 10000$), a size histogram can be derived thereafter, see the upper figure 2. In each diameter bin, a certain number
of droplets are present. Consequently, the volume of liquid water in these bins can be computed and plotted as a histogram, represented in the lower part of figure 2. Note that the peaks of the two different distribution plots are not at the same diameter, since very small droplets only provide a minor contribution to the volume. Having cumulated the volume distribution, the MVD is located at 50% of the total volume.

The choice of boundary conditions for conducting a specific icing wind tunnel experiment depends on the application under investigation. When testing to support industrial product developments, certification requirements are often defining the boundary conditions. Icing of large transport aircraft is addressed i.a. in the document CS25 (EASA (2016)) of the European Aviation Safety Agency. In its Appendix C, information on the standard LWC-MVD envelope can be found, where the liquid water content is in the range between 0.1 and 2.9 g/m$^3$ and MVD between 15 and 50 $\mu$m. Icing certification for wind energy



converters is governed by the technical standards of the International Electrotechnical Comission (IEC). Among others, the *Det Norske Veritas Germanischer Lloyd* (DNV GL) is an accredited registrar and classification society responsible for certification of wind energy converters, having created calculation rules and detailed specification based on the IEC, see GL (2010).

In many cases, the industrial product is significantly larger than the dimension of a typical icing wind tunnel. Hence, appropriate scaling laws are necessary to overcome the limited ranges of air speed, test section dimension and icing cloud characteristics in icing wind tunnels. The NASA manual of icing scaling methods (Bond and Anderson (2004)) states: "When the full-size model is too large for a given facility, model-size scaling is applied, and when the desired test conditions are outside the facility operating capability, test-condition scaling is required. The objectives of such scaling are usually to insure that the amount of

ice relative to the model size, the shape of the ice accreted, dimensionless ice accretion limits, and the aerodynamic penalties due to the ice are the same as would have been obtained with the desired, or reference, model size or test conditions. It has always been assumed that for adequate scaling, characteristic features, such as glaze horns, need to be simulated in size, location, angle and shape."

Summarizing this introduction, icing is a very challenging field of study that incorporates aspects of meteorology, fluid mechanics, thermodynamics, physics and engineering. Despite the fact that many researchers have been involved in the study of ice accretion over numerous decades, the physics of this phenomenon is still not completely understood. Icing wind tunnels are therefore an essential pillar to advance our knowledge in face of this multidisciplinary challenge. Enduring changes in technical standards or certification processes to continuously improve the safety against icing related incidents further emphasize

the industrial need for icing wind tunnel testing. The cost estimations of the Ice Protection Harmonization Working Group (IPHWG) for aircraft certification with respect to SLD-icing underlines this requirement, see figure 3.

Against this background, the design and construction of the Braunschweig Icing Wind Tunnel began in 2010. It was the goal to contribute a tunnel with sufficiently large dimensions in the test section to support both fundamental and applied icing

research with reasonable low operating costs. During the design, construction and commissioning process, many lessons have been learned, which can not be found in the literature and thus form the outline of this publication. The major components of the Braunschweig Icing Wind Tunnel and their design constraints are presented in chapter 2. The special topic of mixed phase icing, where both supercooled droplets and ice crystals are involved in the icing process, is treated in chapter 3. Together with the international co-authoring partners, the commissioning of the tunnel was realized, which is presented in chapter 4. Finally,

some applications and test results that show the tunnel's capabilities are highlighted, with some concluding remarks on health and safety considerations.



## 2 Design and Construction of the Braunschweig Icing Wind Tunnel

### 2.1 Overall Design

The aim to build a low-budget icing wind tunnel for research purposes entailed major choices on the operational envelope. The main targets in icing wind tunnel design are to obtain a homogeneous distribution of flow velocity, temperature, and a uniform

icing cloud in the test section. Therefore, it is the test section, which governs the design choices.

The first choice was on the dimension of the test section. Addressing customers in aviation, automotive and energy industries requires reasonable aerodynamic testing where the wind-tunnel wall boundary layers are significantly smaller than the size of the test section. In this regard, investigations with Reynolds numbers up to $2 \cdot 10^6$ are a frequent demand. Hence, a cross

section of $0.5 \, \text{m} \times 0.5\text{m}$ was chosen.

The maximum velocity is primarily determined by the power of the wind tunnel drive, see section 2-2.2. However, when considering the heat balance in icing wind tunnels, see section 2-2.3, the importance of the chilling device becomes evident. Figure 4 shows cost estimations for both drive and cooling power, which is based on price lists of leading manufactures in their

respective field. Realizing that the cost for tunnel cooling is about 10 to 20 times more expensive than for the tunnel drive, the maximum velocity in the test section is in direct relation to the available investment costs for building the tunnel. Consistent with the requirement of Reynolds numbers up to $2 \cdot 10^6$, the maximum velocity was chosen to be $40 \frac{\text{m}}{\text{s}}$.

The lower temperature bound inside the test section was set to $-20 \, ^\circ C$. Below this bound, one usually observes rime ice

accretions on the test models. In contrast, glaze ice formation with temperatures closer to $0 \, ^\circ C$ is significantly more complex from the perspective of fundamental research that shall be conducted in the present icing wind tunnel. An overview of the entire tunnel and its adjacent components is given in Figure 46. A comparison with the specifications of selected other icing wind tunnels can be found in table 2.

### 2.2 Tunnel Drive

Given an air temperature of $-20 ^\circ C$, a maximum speed $U_\infty$ and a cross-sectional area $A$ of the tunnel test-section, the required jet power $P_{\text{jet}}$ can be calculated to

$$
\begin{aligned}
P_{\text{jet}} &= \frac{1}{2} \varrho U_\infty^2 \cdot \dot{V}_{\text{air}} \\
&= \frac{1}{2} \varrho U_\infty^2 \cdot (U_\infty \cdot A) = \frac{1}{2} \varrho U_\infty^3 \cdot A \\
&= \frac{1}{2} \cdot 1.38 \frac{kg}{m^3} \cdot \left(40 \frac{m}{s}\right)^3 \cdot 0.25 m^2 \\
&\approx 11 kW \quad .
\end{aligned}
$$



In wind-tunnel design, a power factor $\lambda$ is introduced, which indicates the ratio of fan power $P_{\mathrm{fan}}$ and jet power $P_{\mathrm{jet}}$. For a conventional, closed-loop, low-speed wind tunnel, a value of $\lambda \approx 1.5$ can be assumed. The fan power has to compensate pressure and skin friction losses of the individual tunnel components. High losses are caused by the wall boundary layers of the test section and the tunnel diffuser. Furthermore, the first and the second corner after the diffuser create significant losses. Since the vanes of the first corner are subject to ice accretion, their losses are tremendously higher compared to conventional closed-loop tunnels. Consequently, the power factor was dimensioned to a large value of $\lambda = 2.3$, yielding a fan power of $P_{\mathrm{fan}} \approx 25kW$. An axial fan with an efficiency $\eta_{\mathrm{fan}} \approx 67\%$ was selected for the tunnel drive, the electrical power input is therefore $37kW$. Using a frequency converter, the speed of the three-phase asynchronous motor of the fan drive can be controlled, allowing variable tunnel speeds from $5$ to $40m/s$.

## 2.3 Chilling Device

To adjust and maintain a constant temperature of the air inside the icing wind tunnel, which is well below the freezing point of water, a chilling device is necessary, whose cooling capacity has to exceed the power of the wind tunnel drive.

### 2.3.1 Power Estimation

There are three major heat sources that have to be compensated by the chilling device: the power of the tunnel drive, the heat input through the wind-tunnel walls, and the heat transfer of the water spray. The latter is composed of the *sensible heat* $\dot{Q}_{\mathrm{sensible}}$ to super-cool the water spray from the temperature at which it leaves the pneumatic atomizer (about $20°C$) down to the air temperature inside the tunnel test-section and the *latent heat* $\dot{Q}_{\mathrm{latent}}$. As soon as the supercooled water droplet impacts on a solid substrate, the drop solidifies due to the heterogeneous nucleation and releases the latent heat. Both sensible and latent heat are a function of the liquid water content (LWC)

$$\dot{m}_{\mathrm{water}} = \mathrm{LWC} \cdot \dot{V}_{\mathrm{air}}$$
$$= \mathrm{LWC} \cdot (U_\infty \cdot A)$$
$$\dot{Q}_{\mathrm{sensible}} = \dot{m}_{\mathrm{water}} \cdot c_{\mathrm{sensible}} \cdot \Delta T$$
$$\dot{Q}_{\mathrm{latent}} = \dot{m}_{\mathrm{water}} \cdot c_{\mathrm{latent}} \quad ,$$

where $\dot{m}_{\mathrm{water}}$ is the water mass flow rate of the spray atomizers and $\Delta T$ the temperature difference, with the values for demineralized water

$$c_{\mathrm{sensible}} = 4.183 \frac{kJ}{kg \cdot K}$$
$$c_{\mathrm{latent}} = 334 \frac{kJ}{kg} \quad .$$





For typical operational conditions of the icing wind tunnel, i.e. $U_\infty = 40 m/s$, $\Delta T = 30 K$, and approximations for the fan power and the heat input through the wind-tunnel walls, the necessary cooling capacity can be estimated, see figure 5. When LWC is in the range of $1 g/m^3$, a continuous cooling power of $30 kW$ is necessary. To allow for a five-minute peak load of LWC = $3 g/m^3$ and static tunnel temperatures down to $-20°C$, the system was dimensioned for a maximum cooling power of
$80 kW$.

### 2.3.2   Refrigeration Unit

To provide the icing wind tunnel with 30 kW continuous cooling capacity and 80 kW maximum cooling capacity, a customized cooling unit was built, see the upper right part of figure 46. Core of the system is a refrigeration unit ④ that cools the heat transfer fluid (Therminol® D12) down to a temperature of $-32°C$. To compensate the volume change of the fluid, a surge
tank ④ filled with nitrogen is installed. The cold fluid is then stored in a $4 m^3$-sized buffer tank ⑤, which acts as a hydraulic switch. Using a pump and electrically operated valves ⑥, the volume flow from the buffer tank to the heat exchanger in the wind tunnel is controlled. Depending on this flow rate, the continuous cooling power of 30 kW of the refrigeration unit can be well exceeded. For a 80kW peak load operation, the complete buffer tank is first cooled to $-32°C$, and afterwards its cold heat transfer fluid is pumped through the heat exchanger at the wind tunnel within a short period of 7 minutes. The recovered waste
heat from the cooling process is fed into the regenerative heating system of the institute building. Because of its cold-resistant properties, the piping of the cooling system is made of stainless steel 1.4541, and then insulated with a 4 cm thick insulation layer made of Kaiflex®. The entire system is controlled by PLC.

### 2.3.3   Heat Exchanger

The heat transfer between the Therminol® D12 fluid and the circulating wind tunnel air takes place trough two consecutive
heat exchangers, see figure 6. While meandering from the bottom to the top of the first heat exchanger, the Therminol® D12 fluid increases in temperature. With only one heat exchanger, this would yield an undesired vertical temperature gradient in the passing air flow. The second compartment, in which the Therminol® D12 fluid meanders the opposite way, thus compensates the vertical gradient in air temperature. The entire assembly is installed in front of the third corner in the wind tunnel so that the large distance to the test section promotes a homogenization of the temperature distribution over the tunnel cross-section
through turbulent effects.

Mechanically, the heat exchangers have a cross-sectional area of $1.6 \times 1.6 \, m^2$ and are composed of elliptical tubes made of stainless steel and rectangular, smooth steel fins, which are firmly connected to each other by hot galvanizing. The fin pitch is 3.5 mm. To protect the galvanized steel surface against corrosion, it is coated with ZACOSIN® 2000Q, an epoxy resin-based
protective coating that provides high thermal conductivity by embedded aluminum particles, see also chapter 2-2.8. Alternatively, the heat exchanger could also have been made entirely of stainless steel. Compared to standard steel, however, stainless steel has a significantly lower thermal conductivity (50 W / m / K compared to 15 W / m / K), which is why the surface of the heat exchanger would have to be much larger.





## 2.4 Settling Chamber

To obtain a low turbulence level inside the test section of the tunnel, a settling chamber with honeycomb and screens is introduced. An influence on the longitudinal (parallel to the main flow direction) and lateral flow structures (perpendicular to the main flow direction) has to be distinguished (Scheiman and Brooks (1981)). The honeycomb reduces both the lateral

components of mean velocity and of the larger turbulent eddies, where as the screens reduce the longitudinal components of turbulence or mean-velocity variations across the sectional area. For the Braunschweig Icing Wind Tunnel, a combination of screen - honeycomb - screen - screen was selected. The screens have a mesh size of $6mm$, the honeycomb diameter is $10mm$, its length is $100mm$.

## 2.5 Spray System

A spray system is required to produce a uniform drop distribution in the tunnel test section. In the past five years, two different spray systems have been developed; one system suited for large droplets (MVD $\approx 80\mu$m) and a high liquid water content ($1.1\text{g/m}^3 < \text{LWC} < 3\text{g/m}^3$) and a second spray system for low MVD and low LWC.

The spray system for large droplets is composed by a grid of $5 \times 5$ air-assisted atomizers. Inside one atomizer, a thin jet

of water is destabilized by the shear forces of a transversely directed air stream and finally breaks down into small droplets. The diameter of the water jet, which is in the range of $100\mu$m, determines the water volume flux as well as the drop size. The higher the applied air pressure, the higher the aerodynamic shear forces, and the smaller the drop size. In order to enable a high variability of the drop size distribution for basic tests, the atomizers are designed in a modular manner, that is, the water jet and the air atomizer cap can be exchanged separately.

Stainless steel has been chosen as a material for the atomizers, since otherwise conventional atomizers made of nickel-plated brass are prone to wear due to frequent temperature changes inside the icing tunnel (from $+20\,^{\circ}$C to $-20\,^{\circ}$C). However, the poor thermal conductivity of stainless steel (15 W/m/K) compared to brass (120 W/m/K) has to be considered. Since the spray atomizers are directly exposed to the cold tunnel air, they would freeze without corresponding countermeasures. Therefore, a

heating coil is installed close-by.

The second spray system was designed for low liquid water contents, see figure 7. To achieve a LWC of $0.1\text{g/m}$, a water





volume flux can be estimated to

$$\dot{V}_{\mathrm{H_2O}} = \frac{\mathrm{LWC} \cdot \dot{V}_{\mathrm{air}}}{\varrho_{\mathrm{H_2O}}} = \frac{\mathrm{LWC} \cdot A_{\text{test section}} \cdot U_\infty}{\varrho_{\mathrm{H_2O}}}$$

$$= \frac{0.1\,\frac{\mathrm{g}}{\mathrm{m}^3} \cdot 0.25\,\mathrm{m}^2 \cdot 40\,\frac{\mathrm{m}}{\mathrm{s}}}{1000\,\frac{\mathrm{kg}}{\mathrm{m}^3}}$$

$$= 3.6\,\frac{\mathrm{l}}{\mathrm{h}} \quad . \tag{1}$$

5   Using a spray matrix of $5 \times 5$ atomizers, this yields a volume flux of $0.14\,\frac{1}{\mathrm{h}}$ per atomizer, which is ten times lower than the typical minimum flow rate of air-assisted atomizers in industrial environments. In consequence, electrically-actuated air-assisted atomizers have been chosen for the spray system. The atomizers are pulsating up to 10000 times a minute, making the spray appear to be constant. By changing the cycling frequency and the duty cycle, the LWC can be adjusted without modifying the pressure of water and air supply, which is a major advantage compared to classical air-assisted atomizers.

Water and air are conditioned for the operation of the air-assisted atomizers, see figure 8. A pressurized vessel of stainless steel serves as a reservoir of water, whose temperature can be regulated between $2\,^\circ\mathrm{C}$ and $80\,^\circ\mathrm{C}$. Water pressure and air pressure can be adjusted separately between 0.5 and 9 bar. Water flow rate and air flow rate are measured and adjusted by flow regulators. Possible pollutant particles, which could block the nozzles, are separated with water filters.

15   **2.6   Tunnel Nozzle Design**

Having passed the settling chamber, the flow is accelerated in the tunnel nozzle that contracts towards the dimensions of the test section. Beyond altering the turbulence structure (Uberoi (1956)), the shape of the tunnel nozzle also influences the trajectories of the water droplets that are injected by the spray system.

20   Two different nozzle geometries have been considered. These are

- the AVA nozzle, with its contour given by

$$\frac{z\,(x)}{z_{\text{out}}} = \left(\sqrt{K} - 1\right)\left(1 - \frac{x}{L}\right)^3 \left[2 - \left(1 - \frac{x}{L}\right)^3\right] + 1\,, \text{ and}$$

- the Witoszinski nozzle, with its contour given by

$$\frac{z\,(x)}{z_{\text{out}}} = \frac{1}{\sqrt{1 + \left(\frac{1}{K} - 1\right)\frac{\left[1 - \left(\frac{x}{L}\right)^2\right]^2}{\left[1 + \frac{1}{3}\left(\frac{x}{L}\right)^2\right]^3}}}\,,$$





where $x, z$ are spatial coordinates, $L$ is the nozzle length, $z_{\text{in}}, z_{\text{out}}$ are half of the inlet and outlet diameter of the nozzle, and $K$ is given by $K = \left( \frac{z_{\text{in}}}{z_{\text{out}}} \right)^2$. To provide enough time for the super-cooling process of the droplets on their way to the tunnel test section, a large nozzle length of 3.5 m is foreseen.

The droplet trajectories are mainly governed by inertial forces, drag and gravity. A water droplet with diameter $d$, velocity vector $\begin{pmatrix} u_d \\ w_d \end{pmatrix}$ is transported by the air velocity $\begin{pmatrix} u_{\text{air}} \\ w_{\text{air}} \end{pmatrix}$ resulting in a drag force

$$\boldsymbol{D} = \frac{\varrho_{\text{air}}}{2} \left| \Delta \boldsymbol{u} \right|^2 d^2 c_d \cdot \frac{\Delta \boldsymbol{u}}{|\Delta \boldsymbol{u}|} \quad ,$$

where $\Delta \boldsymbol{u} = \begin{pmatrix} u_{\text{air}} \\ w_{\text{air}} \end{pmatrix} - \begin{pmatrix} u_d \\ w_d \end{pmatrix}$ represents the slip velocity. Assuming laminar flow around the spherical droplets with a mass of $m_d$ at low Reynolds numbers $\text{Re} = \frac{\varrho_{\text{air}} \cdot |\Delta \boldsymbol{u}| \cdot d}{\mu_{\text{air}}}$, the drag coefficient $c_d$ can be approximated to $c_d = \frac{6\pi}{\text{Re}}$. Applying the principle of

d'Alembert, this yields a system of differential equations, given by

$$m_d \frac{\partial \begin{pmatrix} u_d \\ w_d \end{pmatrix}}{\partial t} = \boldsymbol{D} + m_d \boldsymbol{g} \quad ,$$

where $t$ represents the time and $\boldsymbol{g}$ is the acceleration due to gravity.

The effect of gravity on the droplet trajectories is depicted in figure 9, assuming an air velocity of 40 $\frac{m}{s}$ at the nozzle exit.
Small droplets with a size of 60 $\mu$m well follow the streamlines of air flow inside the nozzle. With increasing droplet size, the gravitational deflection becomes predominant, particularly at the low speed regions of the nozzle. For 300 $\mu$m droplets, their trajectory from the lowest spray-bar even collides with the nozzle wall. Such undesired collisions alter the size distribution of the droplet cloud and have to be avoided.

Figure 10 demonstrates the effect of the nozzle geometry on the droplet slip velocity. Note that after the nozzle length of $L = 3.5$ m, a straight segment of 0.5 m in length is attached. In the low speed region $x < 1.5$ m, the AVA nozzle accelerates the flow less than the Witoszinski nozzle. Consequently, the slip in velocity between droplet and air is larger in that region. Downstream, for $x > 3$ m, the slip is caught up. While the small droplet of 60 $\mu$m approaches the air speed of 40 $\frac{m}{s}$ when entering the test section at $x = 4$ m, the large droplet of 300 $\mu$m still has a slip of about 5 $\frac{m}{s}$.

Furthermore, the cooling process of the droplet on its trajectory has to be considered. Cooling is promoted by a convective heat transfer at the interface between droplet and air. When the droplet is smaller than 100 $\mu$m, the surface temperature penetrates the entire droplet volume in about 10 milliseconds. Hence, isothermal cooling can be assumed as an approximation for





the thermal energy balance, yielding a differential equation for the droplet temperature $T_d$

$$\frac{\pi}{6}d^3 \varrho_d c_{\text{p, d}} \frac{\partial T_d}{\partial t} = \text{Nu} \cdot \pi d^2 \left(T_{\text{air}} - T_d\right) \cdot \frac{k_{\text{air}}}{d} \quad ,$$

where $\varrho_d$ and $c_{\text{p, d}}$ represent the density and the specific heat capacity of the droplet, $k_{\text{air}}$ is the heat conductivity of air, and Nu is the Nusselt number, estimated by $\text{Nu} = 2 + 0.6\,\text{Pr}^{\frac{1}{3}}\,\text{Re}^{\frac{1}{2}}$, see Knudsen and Katz (1958). Herein, $\text{Pr} = 0.7$ is the Prandtl

number of air.

Using the above equation, the temperature evolution of the droplet inside the tunnel nozzle is plotted in figure 10 for a given static air temperature of $T_{\text{air}} = 263.15$ K. After one meter traveling distance, the droplet of 60 $\mu$m, which had an initial temperature of $T_{\text{air}} = 293.15$ K, has already reached the air temperature. In contrast, the droplet of 300 $\mu$m has still a temperature

above the freezing point of water, when entering the test section at $x = 4$ m.

In summary, droplet conditions up to a maximum size of 150 $\mu$m can be well provided in the present tunnel environment, tanking into account a reasonably low deflection due to gravity, a low slip velocity and sufficient droplet cooling.

## 2.7   Tunnel Walls

### 2.7.1   Mechanical Design

To minimize the heat transport through the wind tunnel walls, they consist of three different layers with a total thickness of 80 mm, see figure 11. Between two layers of aluminum, an insulating foam with low heat conductivity is introduced. The inner aluminum shell which is potentially exposed to the icing cloud is anodized for a protection against corrosion. The flange of each wall segment is made of purenit®, a highly compressed material based on PUR/PIR rigid foam providing a high thermal

insulation. A rubber seal prevents the water from penetrating into the wall.

### 2.7.2   Aerodynamic Wall Effects

Icing wind tunnel tests usually involve models with large dimensions to minimize uncertainties when applying scaling laws for ice accretion. The flow around these models thus interferes with the walls of the test section. Two different wall effects have to be considered. On the one hand this is the viscous effect of junction flow (Simpson (2001)), which occurs when the boundary

layer of the wind tunnel wall encounters the test model attached to the same wall. The adverse pressure gradient forces the wall boundary layer to separate with horseshoe vortices that wrap around the obstacle. These unsteady vortices result in high turbulence intensities, high surface pressure fluctuations and heat transfer rates, which finally alter the ice accretion in that area. This effect was computationally studied with the icing code FENSAP (Beaugendre et al. (2003)), see figure 12. FENSAP solves the Reynolds-averaged Navier-Stokes equations using a finite element approach, calculates droplet trajectories with an

Eulerian scheme and finally determines the ice accretion using a model of Messinger (1953). The computed shear stress, plot-



ted in gray scale, clearly identifies the horseshoe vortex at the wall junction. In its vicinity, the ice accretion is altered, however, the two-dimensional ice shape distant to the tunnel wall, which covers 80% of the wing span, remains unaffected.

The second case of wind-tunnel wall interaction is an inviscid effect. The distant walls that are not connected to the test
model will influence the streamlines around the model. Thereby, both pressure and shear stress distribution along the model surface are altered. Interestingly, at low speeds and sufficiently large droplet sizes, the ice accretion is nearly unaffected, when tunnel walls are introduced, see the computational results of the icing code TAUICE in figure 13.

## 2.8 Corrosion

Since the operation of the icing wind tunnel entails the usage of water droplets or ice crystals, corrosion has to be considered
during the design process. The most vulnerable part of the tunnel in this regard is the heat exchanger, as it is made from galvanized mild steel, see chapter 2-2.3-2.3.3. Many factors decide on the rate of corrosion of a galvanized surface, the most important for this application being the chemical composition of the water and the type of surface contact.

Water with a high content of carbonates supports the creation of a protective layer of alkaline zinc carbonates $ZnCO_3$ on
the galvanized surface. However, with no or only few carbonates and a relatively high content of oxygen, the water reacts with the zinc to form zinc hydroxide $2\,Zn + 2\,H_2O + O_2 = 2\,Zn\,(OH)_2$. This further forms a compound $2\,ZnCO_3 \cdot 3\,Zn\,(OH)_2$, also known as *white rust*, which does not have the protective function of zinc-carbonates, but only connects loosely to the surface. Similar to normal rust this porous layer keeps water near the surface and delays drying, which further increases the rate of corrosion. Over time, particles flake off, until the zinc is used up.

The type of surface contact between water and the galvanized surface is important for the rate of corrosion. If the water is sprayed onto the surface and remains as small droplets, each droplet forms a small corrosion element with a large surface that promotes the supply of oxygen. This increases the corrosion rate in comparison to the case, where a galvanized surface is completely submerged in water with only a small contact area to the surrounding air.

To protect the galvanized steel surface against corrosion, it is coated with ZACOSIN® 2000Q, an epoxy resin-based protective coating that provides high thermal conductivity by embedded aluminum particles.

Another metal in contact with the deionized water is brass, which is used in several valves, pipe connections, etc. Brass
30    forms tarnish, a thin dark layer of stable metal oxides, that protects the base metal. Unlike rust, the oxidation reaction is self limiting. As soon as the layer is formed, no more metal is oxidized. The tarnish did not pose any problem in all parts.

The models for the tunnel are made from aluminum or glass-fibre reinforced plastics (GRP). None of these materials showed



any signs of reaction with the water so far. The same holds for the inner wall of the wind tunnel which is made of anodized aluminum. Even with scratches from installing models no signs of corrosion showed up.

## 2.9 Basic Instrumentation

To monitor the icing wind tunnel operation and thus the boundary conditions of an experiment, several probes are necessary, see figure 14. The velocity $U_\infty$ in the test section is determined by two static pressure probes $p_1$ and $p_2$. Using the Bernoulli-equation

$$p_1 + \frac{\varrho}{2} v_1^2 = p_2 + \frac{\varrho}{2} U_\infty^2 + \zeta_{\text{nozzle}} \frac{\varrho}{2} U_\infty^2 \quad ,$$

where $v_1$ is the velocity in the settling chamber at $p_1$, $\varrho$ is the density of air, $\zeta_{\text{nozzle}}$ is the specific loss coefficient of the wind tunnel nozzle, and a continuity equation

$$v_1 \cdot A_1 = U_\infty \cdot A_\infty,$$

where $A_1$ and $A_\infty$ are the cross sectional areas of settling chamber and test section, one can estimate

$$U_\infty = \frac{p_2 - p_1}{\frac{\varrho}{2} \left[ \left( \frac{A_\infty}{A_1} \right)^2 - 1 - \zeta_{\text{nozzle}} \right]} \quad . \tag{2}$$

Note that the effect of pressure losses in the wind tunnel nozzle is usually neglected, since $\left( \frac{A_\infty}{A_1} \right)^2 \gg 1$ and $\zeta_{\text{nozzle}} \ll 1$. A VAISALA HMT-337 monitors both the relative humidity RH and the total temperature $T_{\text{tot}}$ in the settling chamber. $T_{\text{tot}}$ is used to calculate the static temperature $T_\infty$ in the test section

$$T_\infty = T_{\text{tot}} - \frac{U_\infty^2}{2 c_p} \quad ,$$

where $c_p$ is the specific heat capacity of air. Moreover, the temperature information is necessary to determine the density $\varrho$ of air in equation (2). The power of the tunnel drive $P_{\text{fan}}$ and the pressure loss over the heat exchanger $\Delta p_{\text{HE}}$ indicate the performance degradation of the icing wind tunnel due to ice accretion. Furthermore, the state of the spray system $\mathcal{S}_{\text{spray}}$ is controlled, see section 2-2.5.

## 3 Ice Crystals and Mixed Phase Capability

Ice crystals in the atmosphere have to be considered for aircraft safety, since they partially melt in warm environments and develop a "sticky" character. In particular, heated stagnation pressure probes and engine compressor stator blades can be mal-effected by ice crystal icing (Mason et al. (2006)). Icing of aircraft probes can cause false flight parameters displayed inside the





cockpit. Ice accretion inside the compressor causes flow blockage, forcing the compressor to operate towards stall conditions. The compressor encounters a decay in rotational speed resulting in significant thrust losses (rollback event) (Oliver (2014)). Moreover, total engine flame out may appear if a huge mass of accumulated ice is shed into the combustor.

To provide experimental capability on ice crystal icing, the Braunschweig icing wind tunnel was upgraded with an ice crystal generation and conveyance system, which is presented in this section, see also figure 46.

### 3.1 Morphology of Ice Crystals

Ice crystal icing conditions are typically encountered in wide ranging convective cloud systems of high ice water content at flight altitude. Such conditions can especially be found in the vicinity of mesoscale convective cloud systems in tropical re-
10 gions, see Grzych and Mason (2010) and Leroy et al. (2015). In order to better document ice crystal icing conditions, two flight campaigns have been conducted in the course of the HAIC- and HIWC project (Dezitter et al. (2013), Strapp et al. (2016)). The first campaign took place in Darwin, Australia in 2014 during the monsoon period, the second campaign in Cayenne, French Guiana, in 2015 during the rainy season. The flight measurements have been conducted in high ice water content cloud areas in large tropical mesoscale convective systems, mostly over the oceans. Details about the campaigns and about data treatment
can be found in Leroy et al. (2016b), Leroy et al. (2016a) and Leroy et al. (2017).

Atmospheric ice crystals feature a broad diversity of sizes and shapes which depend on the individual ice crystal growth history affected by ambient temperature and super-saturation. Figure 15 and 16 show examples of ice particle images captured close to $-10\,°C$ during the Darwin campaign with the 2D-Stereo probe (see section 4-4.5 for a description of the instrument).
Images from figure 15 correspond to a stratiform part of the cloud where rather constant ice water contents close to $1.0\,g/m^3$ were sampled. Images from figure 16 were recorded in convective cores with IWC peak values exceeding $3\,g/m^3$. In the convective part, ice crystals are more numerous and smaller. Close to $-10\,°C$, column and capped column type crystals have been found. Larger ice crystals ($> 600\,\mu m$) are rare and resemble graupel (dense and roundish particles). On the contrary, in the stratiform part of the cloud, ice crystals larger than $600\,\mu m$ are more frequent and appear to be less fragile, consisting of
aggregates of pristine shapes. Since particle growth is affected by vapor deposition and aggregation and encounters different temperatures regimes, a lot of different and irregular shapes are possible at all altitudes in mesoscale convective cloud systems.

Figure 17 shows ice particle size and mass distributions (PSD and MSD) depending on ambient temperature using the equivalent diameter $d_{eq}$ for size definition. The size and mass distributions have been averaged for the selected temperature
regimes, only cloud areas with total water contents above $1.0\,g/m^3$ have been taken into account. The concentrations of small ($< 200\,\mu m$) and large ($> 1\,mm$) ice crystals vary in opposite ways with temperature: For colder temperatures, concentrations of small ice crystals increase whereas the number of large crystals decreases (cf. figure 17). This temperature dependency might be a consequence of several cloud processes. Nucleation of new ice particles is favored at low temperatures, creating new small ice crystals. On the opposite, growing of ice crystals by collection processes requires larger particle sizes and might





be more efficient at higher temperature. Regarding the dynamics of the cloud, small ice crystals can also easily be carried aloft by updraft winds. Therefore, there is still no clear and unique scenario describing the formation of high IWC areas in clouds. Cumulative ice particle mass distributions are plotted in the bottom part of figure 17. PSDs have been converted in to MSDs following the work of Fontaine et al. (2014) and Leroy et al. (2016b). Bin masses are linked to the particle size using a

5 power-law relationship $m(D) = \alpha D_{eq}^{\beta}$. However, $\alpha$ and $\beta$ are not constant; for each time step, $\beta$ is deduced from the analysis of particle?s images and $\alpha$ is constrained by additional TWC measurements thus ensuring that the total mass from the MSDs equals the measured TWC. With decreasing atmospheric temperature the cumulative particle mass is carried more and more by small ice crystals. The median mass diameter (MMD), reduces from roughly 750 $\mu m$ at $-10\,°C$ to 320 $\mu m$ at $-50\,°C$. More details about MMDs in high IWC cloud regions can be found in Leroy et al. (2017).

As mentioned above, ice crystal icing can cause malfunctions of aircraft engines due to inner ice accretions. Inside-engine conditions are characterized by ice particle sizes of about 20 $\mu m$ as ice crystals fragment in the fan stage. Due to centrifugal forces, high ice particle concentrations can be found in the casing region of the core engine. The local TWC can exceed the atmospheric TWC by a factor of 4 or even more, see Feulner et al. (2015).

## 3.2 Ice Crystal Production in a Cloud Chamber

It has been aimed to reproduce closest possible replicates of natural ice crystals. Cloud chamber technology has been identified to be appropriate, because natural ice crystal growth is simulated in an artificial cloud. Usually, cloud chamber technology is applied in meteorological science to investigate ice crystal formation and growth mechanisms and to study the interaction of individual ice crystals, see Connolly et al. (2011). Thus, the productivity of cloud chambers is accepted to be rather low. For

icing wind tunnel studies, huge amounts of ice particles are required. In collaboration with the Austrian Neuschnee GmbH, two highly productive cloud chambers have been developed and installed inside a cooling room. Figure 18 and 46 show both cloud chambers and auxiliary equipment. Basically, atomized droplets are forced to freeze out inside the chamber by the expansion of cool pressurized air. Strong circulation of air keeps the particles in suspense until they grow to certain size and settle down to the bottom of the chamber. The production rate of the cloud chamber is not sufficient to enable a direct supply of ice crystals

into the wind tunnel. Thus, the particles are collected inside a chest freezer, which is directly connected to the cloud chambers. The chest freezers operate at -70 $°C$ to minimize degeneration and sintering of the particles. Currently, the production rate for both cloud chambers is limited to about 1kg/h. This production rate is sufficient to perform between 10 to 15 wind tunnel tests a day.

Microscopic images of ice crystals grown inside the cloud chamber are shown in figure 19. The primary ice particle habit can be adjusted by variation of ambient temperature inside the cooling chamber. After storing in the chest freezer, the ice particles typically feature aggregates of individual crystals as further illustrated in the following section.



### 3.3 Ice Crystal Conveyance

To establish defined cloud conditions inside the IWT test section, ice particles are fluidized into an airflow and guided into the wind tunnel, see figure 46. The conveying airflow is extracted from the icing wind tunnel by a bypass construction including a radial fan. An external heat exchanger (aftercooler) cools the air to compensate heat input of the fan and environment. The

aftercooler is connected to the refrigeration system of the cooling chamber. Downstream of the heat exchanger, the piping system enters the cooling room where ice particles are supplied to the airflow. Further downstream, the particle-laden flow is discharged into the IWT.

To adjust the ice water content inside the test section a defined massflow of ice particles $\dot{m}_{ice,inj}$ has to be supplied at a

constant feed rate. Particle dosing is realized by a volumetric dosing machine, shown in figure 20. The dosing machine has a very linear operating behavior: the higher the frequency of the dosing machine motor, the higher the supplied ice particle massflow, see figure 21 (left). After storage and dosing, the particles are partially interlocked and exist as accumulated ice clumps, which are by far larger than the desired particle sizes of several tens of microns. Thus, the particles have to be sieved to desired sizes, which is realized by a custom made sieving machine, see again figure 20. The sieving machine is driven by an

electric motor that forces a 800x800 mm sieve to oscillate at a frequency of 8 Hz. The total mass of ice provided by the dosing machine does not pass the sieving machine. Greater clumps of ice are not fully broken by the sieving procedure and depose on the sieve. The average deposit is about 15 to 20 percent of the ice mass provided by the sieving machine.

The sieving machine is connected to the conveyance piping system by a conical flexible bag. Ice particles passing the sieve

directly fall into the piping system and get dragged by the pipe airflow. To adapt the local pressure inside the piping system to the ambient pressure inside the cooling room, the piping system implies an injector nozzle that induces a local jet inside the pipe, see the lower part of figure 20. The nozzle diameter is carefully adjusted based on preliminary calculations of the pressure distribution inside the whole piping system. The temporal stability of the mass flow is depicted in figure 21 (right). The dashed line shows the ice particle mass supplied by the dosing machine at a frequency of 21 Hz. It is a linear fit, a constant feedrate

has been proven by weighting, as indicated by the red data points. The blue dots and the solid blue line represents masses of ice weighted at the outlet of the sieving machine. This curve shows a very linear behavior as well. One can conclude that after about 20 seconds the ice particle massflow injected into the wind tunnel is temporally constant. Weighing of ice mass passed by the sieving machine has been repeated several times, also for lower feedrates and it turned out that after about 30 seconds the provided ice massflow is stable. Correlations between the injected ice particle massflow $\dot{m}_{ice,inj}$ and the ice water content

$IWC$ inside the test section are discussed in chapter 4-4.6. Figure 22 shows a microscopic image of two ice particles which have been captured at the exit of the sieving machine. As one can see, the particles feature aggregates of tiny ice crystals and can thus be considered as close replicates to the natural ice crystals shown in figure 15 and 16.

The ice particles are injected into the settling chamber of the icing wind tunnel, upstream of the spraybar system, see fig-



ure 23. The velocity of the particle-laden jet is about 20 m/s, which is 5 times higher than the local wind tunnel airspeed. Consequently, the jet mixes and expands by turbulent mixing with the ambient airflow so that the particles are spread among the flow field and mix with the droplets atomized by the spraybar system. The particle trajectories get contracted inside the wind tunnel nozzle. Due to the circular pipe exit the ice particles also cover a circular cross sectional area inside the test sec-

tion. It has been tried to extend this area by using adapters at the pipe exit and by the use of several injection pipes distributed along the settling chamber. These efforts did not work out due to ice deposits inside the additional assembly, see Baumert et al. (2016). Consequently, the simplest approach of the single pipe outlet has been chosen. For studies at glaciated and mixed phase conditions, screens and honeycombs in the settling chamber of the tunnel have to be unmounted to avoid ice accretion on these elements.

# 4 Commissioning of the Braunschweig Icing Tunnel

## 4.1 General Approach

The icing-wind-tunnel calibration has been performed with respect to the requirements specified in SAE ARP 5905. Accordingly, both an aero-thermal calibration of the airflow and a calibration of the icing cloud have to be performed. After presenting the deployed measurement techniques, selected calibration results are presented. Finally, an instrumentation inter-comparison

exercise was carried out.

## 4.2 PDI Probe

The measurement principle of PDI is based on light scattering interferometry (Rudoff et al. (1990)) that uses as measurement scale the wavelength of light and as such its calibration is not as easily degraded. The scattering by spheres much larger than the instrument wavelength is approximated by geometrical optics. Size and velocity are determined by measuring sinusoidal

scattering signals on adjacent photo-detectors as particles move through an interference fringe pattern formed in the intersection of two laser beams of same wavelength.

The method does not require frequent calibration since the light wavelength does not change and the detector separations that affect the size measurement are fixed. The sinusoidal nature of the signals detected may be used with the Fourier analysis

approach to detect signals reliably even in low signal-to-noise (SNR) environments. Off-axis light scatter detection is used. The advantage of this approach is that a very small, well-defined sample volume can be formed which minimizes uncertainty when computing absolute concentration and reduces the possibility of coincident events (more than one particle residing in the sample volume at one time). The sample volume can be adjusted to balance count rate with coincidence rate to suit the user?s preference. The PDI only needs an initial factory calibration since parameters affecting the measurements as the laser

wavelength, beam intersection angle, transmitter and receiver focal lengths, and the detector separation do not change with age or use of the instrument.





The method has been demonstrated to be capable of measuring drop size distributions, droplet velocity distributions, size-velocity correlation, droplet time of arrival, droplet spacing, droplet number density, liquid volume flux, and liquid water content.

## 4.3 Cranfield Isokinetic Probe (IKP)

The probe consists of two tubes, where one pipe end is faced towards the upstream direction, thereby collecting a representative sample of air, droplets and ice crystals suspended within the flow, see also figure 24. The goal is to measure the total water content TWC of the flow. The volume flux $\dot{Q}_{\text{front}}$ has to be adjusted to obtain isokineticity at the probe head. The second tube is faced downstream with its pipe end. Thus, a second sample of air, without any particles or droplets in it, can be collected to determine the water vapour content so that the quantity of water in condensed form (solid and liquid) can be determined.

The probe stays free of an ice build-up so that the flow conditions at the entrance to the probe can be carefully matched to oncoming flow in order to achieve the correct sampling, not too rich or depleted with respect to the condensed water. The heating of the inside of the probe drives all sampled water droplets and ice crystals into the vapor phase. A subsequent cooling system composed of copper tubes plunged into a water tank has been added in order to bring the air at a reasonable temperature (about 30-35 °C) before entering the measurement system.

The measurement system itself is mainly comprised of two parts. The first part is the mass flow meter measurement and valve to automatically control the mass flow at the probe inlet to maintain iso-kinetic conditions. The second part is the water vapor concentration measurement which is done using a LICOR 7000 system.

The probe features a double wall construction making it possible to use it as a pitot static probe and also providing a means to heat it. A large current is passed through the outer wall, via the tip and back along the inner tube. Tube wall thicknesses are chosen to get the appropriate split of heat between the different parts of the probe.

## 4.4 HSI Probe

The high speed imaging (HSI) probe (Baumert et al. (2016)) illuminates a volume using six laser beams and takes shadow images of traversing cloud particles. Thereby, depth of field errors and sampling bias due to particle obscuration are minimized. The shadow images are sampled at a frequency of 300 fps by a CMOS-chip of 640 by 480 pixels. Images of particles in the range of 5 to 1200 $\mu$m can be recorded, the resolution is 3.795 $\mu$m/pixel. The entire device including lasers and camera is remotely controlled. A trigger beam of different wavelength is coaxially aligned to the laser beams, see figure 25 (left). A receiver including an aperture of appropriate size and shape is elevated to 40 degrees to the transmitted beam and detects particles passing the object plane. It will trigger the lasers (laser pulse duration 12 ns) when particles are within the desired measurement volume. The HSI hardware is integrated in a cylindrical canister. Three arms including optical components are





mounted on the front end of the canister that allow to take intrusive particle images, see also figure 25.

The image processing software is able to detect and to evaluate irregular shaped particles. Some examples of 2D particle images that were detected by the HSI probe, are shown in figure 25 (right). Particles that are rejected by out-of-focus or other
validation criteria are marked in red, valid particles are outlined in green colour.

### 4.5   2DS Probe

The two-dimensional stereo (2D-S) probe (Lawson et al. (2006), Lawson and Baker (2006)) detects the size and concentration of cloud droplets and ice particles in the size range of 10 to 1280 $\mu$m using shadow images of the cloud particles. Two orthogonal diode laser beams illuminate two linear diode arrays consisting of 128 photodiodes with 10 $\mu$m pixel resolution.
When a particle crosses the laser beam in the sampling volume, its shadow image on the photodiode array is recorded by high-speed electronics. The diode lasers operate at 45 W and are single-mode and temperature-stabilized. This design with two lasers better defines the sampling volume boundaries and thus minimizes errors associated with the depths of field and the sizing of small particles.

### 4.6   Calibration Results

#### 4.6.1   Aerothermal Calibration

An aerothermal calibration has been conducted to ensure and prove adequate airflow quality inside the test section. Investigations on airflow uniformity have been made by means of a five-hole-probe. The probe has been calibrated *a-priori* based on the procedure described by Treaster and Yocum (1978). The measurements have been performed for the IWT-standard setup with screens and honeycombs installed in the settling chamber and also for the ice-crystal-icing setup, where these elements have
been unmounted.

Figure 26 shows the contour of axial velocity inside the test section for the ice-crystal-icing setup. The pneumatic system for ice-particle conveyance was activated, the exit velocity of the pipe depicted in figure 23 was about 20 m/s, however, no ice particles were fed into the system. The plot covers a cross-sectional area of $240 \times 400 \mathrm{mm}^2$ around the center-line position.
Measurements have been taken in vertical and horizontal steps of 40 mm. One can clearly observe a footprint of the pneumatic-conveyance jet inside the flow-field. Velocity deviations relative to the center-line velocity are about 0.4 m/s, which complies with ARP 5905 requirements of +/- 1 % spatial flow uniformity. Moreover, vectors of orthogonal velocity components are plotted, indicating a vortex inside the flow field. Since no screens and no honeycombs are installed, the vortex most likely originates from the flow turning in the third and forth corner of the wind tunnel. The vortex magnitude is very low, local flow
angularities comply with ARP 5905 specifications. For the IWT-standard setup, no vortex can observed and the flow field is very uniform as shown in figure 27.





Turbulence intensity has been measured by means of hot-wire anemometry. Figure 28 shows the contour plot of turbulence intensity for the ice-crystal-icing setup. The peak value of 1.4 percent complies with ARP specifications. Again, the footprint of the pneumatic-conveyance jet is clearly visible.

Besides flow velocity characterization, measurements of total temperature have been carried out. A very uniform temperature distribution could be verified for the IWT-standard setup with flow straightener and screens installed. Local temperature deviations during the measurements correspond to temporal variation of tunnel temperature, which can be adjusted with an accuracy of +/- 0.5 $°C$ around the target value. For the ice-crystal-icing setup, a footprint of the jet can be found in the temperature field. For a static wind tunnel temperature of 0 $°C$, the local temperature at center-line position is about $-0.8$ $°C$.

At lower temperatures, the discrepancy diminishes since the jet temperature and the airflow temperature converge. In case of rather low temperatures of $-15$ $°C$, the center-line temperature is about 0.4 K higher than the ambient temperature.

### 4.6.2   Calibration of the droplet cloud

Currently, two spray systems can be mounted inside the IWT, see section 2-2.5. The spray system with high LWC and high MVD has been calibrated by means of the Cranfield Isokinetic Probe (IKP), the canister PDI probe, and the modular HSI probe.

IKP measurements have been conducted to correlate the liquid water content (LWC) in the test section with the water flow-rate that is supplied to the spray atomizers. The left image of figure 29 shows temporal plots of LWC obtained for various water flow-rates at $U_\infty = 40$ m/s. As illustrated in the right image of figure 29, the dependency between LWC and flow-rate is rather linear. Test points corresponding to the plots in the left image are highlighted by the same color. For lower flow velocities $U_\infty$, higher LWCs values are obtained as anticipated in equation 1. Nevertheless, the LWC increase is not proportional to $1/U_\infty$,

because the cross-sectional area of the icing cloud expands at lower wind-tunnel speeds.

Comparisons of liquid water contents derived from canister-HSI and canister-PDI measurements are plotted in the left image of figure 30. The dashed lines stress deviations of +/- 10 percent. The liquid water contents assigned to C-HSI and C-PDI have been derived based on volume to mass correlations of detected droplets. It can be observed that there is very good agreement

between the measurements of all three probes except for to test points of the PDI at $U_\infty = 20\,m/s$. These measurements had been affected by water condensation inside the C-PDI. The MVD varies in the range of 80 $\mu m$ to 95 $\mu m$, see the right image of figure 30. Higher MVDs have been observed with the C-PDI probe at $U_\infty = 40\,m/s$. Presumably, droplet coalescence is promoted for higher air velocities. Yet the medium droplet size is rather constant for various liquid water contents at the same wind tunnel speed. All these results refer to the ice-crystal-icing setup with the jet switched on. No measurements

have been taken away from center-line position as neither the canister probes nor the IKP could be traversed adequately. Ice accretion studies of selected test articles have proved a very uniform droplet cloud.





### 4.6.3    Calibration of the ice particle cloud

Ice-particle cloud calibration has been performed by means of the IKP and the canister-HSI. Figure 31 shows time histories of the ice water content (IWC) determined by the IKP for dosing frequencies $f_{\text{dosing}}$ of 5 and 30 Hz at wind tunnel speeds $U_\infty$ of 20 and 40 m/s. In the left image absolute values are plotted while the right image shows plots of IWC normalized by the target IWC. According to the IKP measurements, stable cloud conditions are established after 20 to 60 seconds, which is surprising since steady state particle supply is established after 20 to 30 seconds. A slow response time of the IKP might be a possible explanation. Increasing the particle supply of the dosing machine to frequencies higher than 30 Hz does not result in higher IWC, because the permeability of the sieving mesh is limited. According to figure 31, a maximum IWC of about $19\ g/m^3$ can be achieved at the center-line position. At $f_{\text{dosing}}$ =5 Hz, a minimum IWC of $3\ g/m^3$ can be adjusted. Lower feed-rates can be obtained by use of a smaller feed pipe of the dosing machine but have not been calibrated with respect to test section IWC. Figure 31 shows oscillations in IWC at low feed-rates because of an unsteady discharge of the dosing machine. This issue has been solved by the use of a rotating rod which supports particle discharge.

Based on the IKP measurements, a linear correlation between IWC and dosing frequency has been determined. Corresponding plots are shown in the left image of figure 32 for wind tunnel speeds of $U_\infty = 20\ m/s$ and $U_\infty = 40\ m/s$. Due to the linear operational behavior of the sieving and dosing machine, a strong linear dependence between IWC and injected ice particle mass $\dot{m}_{ice,inj}$ could be demonstrated as shown in the right image of figure 32. This dependency allows to assess the accuracy of IWC set for a test run. For wind tunnel tests which include ice particle supply, the IWC is adjusted based on the linear fits of figure 32, left. The frequency of the dosing machine is set according to the demanded IWC. After a test run, the ice particle deposit on the sieving machine is weighted. Based on the correlation of dosing frequency and supplied massflow, see figure 21 (left), the ice particle massflow $\dot{m}_{ice,inj}$ can be determined respecting ice deposit on the sieve. Figure 32 then allows to deduce the real IWC which has been established inside the test section. Additional test campaigns where this procedure has been applied, showed an accuracy of IWC adjustments of about +/- 15 %.

Investigations on spatial IWC uniformity have been performed by means of a custom-made particle-collecting tube system. The system allows to measure a relative distribution of the ice particle concentration inside the test section. A contour plot for $U_\infty = 40\ m/s$ is shown in figure 33. Acceptable uniformity of +/- 20 percent according to ARP 5905 is given for a circular area of about 150 mm in diameter. This area has proved to be sufficient and appropriate for valid icing experiments. The peak IWC is located slightly above the center-line position, which has to be respected for the mounting of aerodynamic test models.

### 4.6.4    Probe inter-comparison

Based on the IKP calibration, a cross comparison of various optical array probes has been performed in the Braunschweig Icing Wind Tunnel. The CNRS and DLR 2DS-probes, as well as the C-HSI have been mounted consecutively at the same position inside the test section. Additionally, modular HSI and PDI probes have been installed externally, to assess the repeatability of





the test conditions. Figure 34 gives an overview of the test setup. M-PDI and M-HSI laser beams have been adjusted close to the sampling volume to perform non intrusive measurements in parallel.

Figure 35 presents a comparison of optical array probe measurements at seven glaciated cloud conditions named I01, I02,

5    I03, I05, I06, I09 and I10. Ice water contents have been derived based on the mass-size relation described by Brown and Francis (1995). Ice water contents derived from measurements of both 2DS-probes and the C-HSI probe have been related to the estimated wind tunnel IWC. A rather good agreement between the measurements was proven as most of the measured values agreed with the wind tunnel prescribed IWCs with an accuracy of $\pm 20\%$, which is well within the limits of expected instrument accuracy.

Wind tunnel testing of the 2DS-probes and the C-HSI have generated a great variety of ice particle images. Figure 36 shows examples of ice particles captured by the C-HSI. The aggregate structure which has been observed for ice particles before fluidization and conveyance, see figure 15, could be maintained. Figure 37 shows a series of ice particle images obtained from 2D-S measurements inside the Braunschweig IWT. It can be observed that most of the particles feature an irregular, elongated

shape. Based on the analyses of ice particle images one can conclude that the particles partially break up during conveyance and are reduced to smaller sizes.

A characteristic size distribution of artificial ice particles inside the IWT is shown in figure 38. Furthermore, particle size distributions of atmospheric ice particles (Atm) are included according to figure 17. The artificial ice particle cloud includes

higher concentrations of small ice particles with equivalent diameters below 200 $\mu m$s. The amount of particles smaller than 200 $\mu m$ yielded from the wind tunnel measurements has not been identified during the airborne measurements, even for the coldest temperatures. The high amount of small ice particles is reflected in the cumulative mass distribution shown in figure 39. An MMD of about 79 $\mu m$ was determined for the wind tunnel icing cloud. This value agrees with the initial target of an MMD in the range of 50 to 200 $\mu m$. The HAIC flight campaigns have shown that against previous assumptions natural ice crystal

clouds are characterized by higher MMDs in the range of 300 to 800 $\mu m$ depending on temperature, see figure 39. Yet the high number of small ice crystals with MMDs below 80 $\mu m$ may be of relevance for particle size distributions in young and aged contrails (Voigt et al. (2010), Voigt et al. (2011)).

It can be summarized that the initial design targets of the ice-crystal generation system have been achieved. Irregular ice

particles with close habits to natural ice crystals can be injected into the wind tunnel. Even realistic primary habits can be grown by cloud chamber technology and might be used for single particle impact studies. The ice particle size distribution is characterized by an MMD of about 80 $\mu m$ which is between atmospheric and inside-engine particle sizes. IWC values of 3 to 20 $g/m^3$ correspond to total water contents expected for inside-engine conditions. Lower ice water contents can be adjusted by modifications of the dosing machine but require further wind tunnel calibration measurements.



## 5  Some applications and test results

### 5.1  Super-cooled Large Droplets Icing

The consideration of super-cooled large droplets (SLD) icing is part of the certification for large transport aircraft (Politovich (1989)). Appendix O of the document CS25 (EASA (2016)) of the European Aviation Safety Agency provides detailed in-

formation on the MVD and LWC envelope. Outstanding requirement is the simulation of liquid clouds with median volume diameters larger than 50 $\mu$m. The trajectory of such large droplets show a large deviation from the air streamlines. With their increased inertia, the droplets might be impacting in regions on the aircraft wing, which are not protected with de- or anti-icing systems.

Such SLD ice shapes and their surface roughness where investigated in the Braunschweig Icing Wind Tunnel (Steiner and Bansmer (2016)). In order to determine which wind tunnel conditions to use, they had to be related to the relevant dimensionless parameters influencing ice roughness. These are the dimensionless accumulation parameter $A_c$, given in Bond and Anderson (2004) as

$$A_c = \frac{LWC \cdot U_\infty \cdot t_{acc}}{\rho_{\text{ice}} \cdot \delta} \tag{3}$$

and the stagnation freezing fraction

$$f_0 = f_0\left(p, T, U_\infty, MVD, LWC, c, \delta\right) \tag{4}$$

where $\rho_{\text{ice}}$ represents the density of ice, $\delta$ the diameter of the inscribed circle of the airfoil nose as a reference length, free-stream pressure $p$ and static free-stream temperature $T$.

Within this two-dimensional ($A_c$ and $f_0$) parameter space, a Latin hypercube sampling was performed to define a set of experiments. The resulting ice shapes are shown in figure 40 and can be categorized in two groups. For shapes in the first group, which are marked in blue, the ice thickness was almost constant over some distance from the stagnation point and then a ridgeline appeared. Shapes in the second group were marked in red, and exhibited an intense yet smooth ice accretion at the stagnation line followed by a much rougher zone with a feather structure.

### 5.2  Mixed-Phase-Icing of a Cylinder

Comprehensive investigations on ice accretion of generic test articles at mixed phase conditions have been performed in the recent past (Bansmer and Baumert (2017)). The test articles have been equipped with heat foils to investigate the effect of internal heat conduction on the accretion process.





Figure 41 shows ice shapes on a cylinder model without heat foil operation. A mixed phase cloud with a total water content of $12 g/m^3$ and a melting ratio $m_r$ (defined as LWC/TWC in this setup) of 0.12 has been adjusted. The static temperature has been varied between 0 and -15 $^\circ C$. In contrast to supercooled droplet icing, the ice accretion for mixed-phase icing exhibits a conical shape. The ice accretion process appears to be strongly dependent on air temperature. At constant melting ratio and

5   constant total water content, a decrease in icing severity can be observed for lower temperatures. This effect can be addressed to supercooling of the liquid droplets inside the mixed phase clouds. At lower temperatures enhanced supercooling allows a large amount of liquid to freeze with only a short delay upon impact. Therefore, less liquid is locally available to promote ice particle sticking resulting in a significantly reduced ice accretion. Water imbibition into the supposed porous ice layer seems therefore a necessary condition for increased icing severity.

Figure 42 gives an example of mixed phase icing of the cylinder with activated heat foils. In contrast to figure 41, no cone shaped ice accretion develops as the accretion layer constantly melts due to internal heating. Consequently, the accretion does not adhere to the cylinder surface and gets dragged towards the upper side of the cylinder by shear forces induced by the ambient airflow. Capillary forces prevent the accretion layer from instant shedding. As the accretion layer convects downstream,

further ice accumulates in the flow stagnation area of the cylinder. In consequence, a cohesive ice accretion structure detaches from the cylinder and growths further downstream.

## 5.3   Development of Measurement Techniques

The experimental characterization of multiphase flows in icing wind tunnels can be very challenging and is a fruitful topic for developing measurement techniques as already shown in chapter 4. Here, we want to shortly summarize an optical technique

to measure the 3-D geometry of ice accretion.

Two properties of ice accretion are particularly demanding for optical measurement techniques. On the one hand, the ice surface geometry is very complex and includes both large roughness elements and deep chasms, see again figure 40. On the other hand, glaze ice formations appear transparent in the visible light range, which makes them hard to detect with conven-

tional camera systems (Gong and Bansmer (2015b)). Therefore, we developed a general approach based on mid-infrared (MIR) laser scanning (Gong and Bansmer (2015a)), because MIR radiation penetrates ice and water only within a depth of less than 10 micrometers.

The MIR-line scanning method uses a $CO_2$-laser light sheet, which is first created through proper lenses, and then projected

onto the scene, see figure 43. The ice surface will absorb this light and emit an infrared radiation signature, which is visible as a deformed line. Using a properly calibrated camera, the deformed line can be transformed using a triangulation function yielding 3D-coordinates. While consecutively scanning with the laser light sheet over the object, the entire 3D ice geometry can be reconstructed.



## 6   Health and Safety

Operating the icing wind tunnel with prototype test models requires intervention and human surveillance. In particular, activities inside the icing wind tunnel or in the cooling chamber (see ⑧ in Figure 46) can be very exhausting and have to be considered for health and safety. As depicted in Figure 44, unsafe behavior significantly increases when the ambient wet-bulb
temperature is lower than $10\,^{\circ}C$, although workers are appropriately clothed. Frequent crew change is thus a necessary measure to perform these exhausting activities.

Another safety aspect is water handling with respect to legionella, see figure 45. These bacteria multiply with ideal growth rates between 32 to $42\,^{\circ}C$ and can infect alveolar macrophages, which may result in several lung diseases. The water, which
is to be sprayed inside the icing wind tunnel is increasing this threat since the contaminated droplets smaller than $10\,\mu$m can be deeply inhaled. To minimize this risk, demineralized water is used that is further sterilized with ultraviolet radiation before entering the pneumatic atomizers.

*Competing interests.*   The authors declare that they have no conflict of interest.

*Acknowledgements.*   The presented work was conducted within the framework of the DLR@Uni programme SuLaDI, which is founded
by the Helmholtz Association. The ice crystal icing capability was achieved by funding from the European Union's Seventh Framework Programme in research, technological development and demonstration under grant agreement nᵒ ACP2-GA-2012-314314. The improved spray system was funded by the German Federal Ministry of Economic Affairs and Energy, grant number Drífa-FKZ0325842A. Some of the contributed computational results were obtained thanks to Prof. Habashi (McGill university, Canada) and the funding of the German Research Foundation, DFG grant BA4953/2-1. Christiane Voigt and Tina Jurkat-Witschas have been funded by the Helmholtz society under
contact no W2/W3-060 and the German Science Foundation within DFG SPP 1294 HALO by grant VO1504/4-1 and grant JU 3059/1-1. Figure 4 thanks to Daniel Hattenbach (Greencooltec GmbH), Figure 46 supported by Walter Arnold (Berlin). The authors further express their thanks to all project partners from Europe and overseas.





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



**Table 1.** Langmuir-D distribution of droplet size Langmuir (1961).

| LWC fraction | $\left(\frac{d}{\text{MVD}}\right)^2$ |
|---|---|
| % | - |
| 5 | 0.31 |
| 10 | 0.52 |
| 20 | 0.71 |
| 30 | 1.00 |
| 20 | 1.37 |
| 10 | 1.74 |
| 5 | 2.22 |

**Table 2.** Performance of different icing wind tunnels compared to the Braunschweig Icing Wind Tunnel. $v_{\text{max}}$ represents the maximum air speed in the test section, $P_{\text{chill}}$ is the installed cooling power. Data extracted from Pastor-Barsi et al. (2012), Vecchione and De Matteis (2003), Al-Khalil et al. (1998) and Oleskiw et al. (2001).

| Tunnel Name | Test Section Size | $v_{\text{max}}$ | $P_{\text{fan}}$ | $P_{\text{chill}}$ | $\lambda$ | $\frac{P_{\text{chill}}}{P_{\text{fan}}}$ |
|---|---|---|---|---|---|---|
| | m$^2$ | m/s | kW | kW | - | - |
| NASA IRT | $1.83 \times 2.75$ | 174.3 | 3730 | - | 0.20 | - |
| CIRA | $2.35 \times 1.15$ | 225.0 | 4000 | 6400 | 0.19 | 1.60 |
| Cox | $0.71 \times 1.17$ | 98.3 | 149 | - | 0.27 | - |
| NRC | $0.57 \times 0.57$ | 100.0 | 450 | 420 | 1.99 | 0.93 |
| **Braunschweig** | $0.50 \times 0.50$ | 40.0 | 37 | 80 | 3.35 | 2.16 |

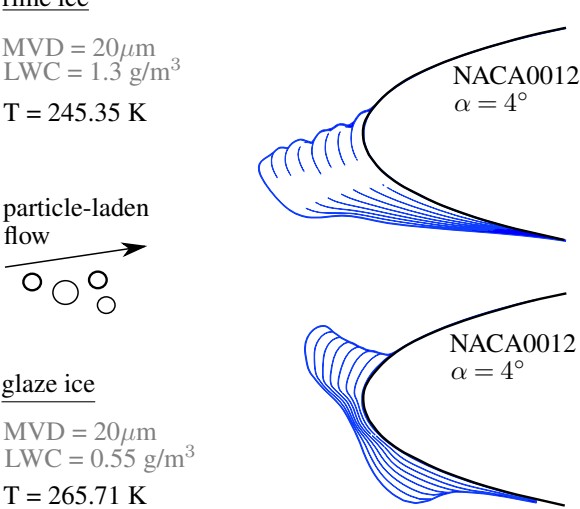

**Figure 1.** Rime ice and glaze ice shapes over time at the leading edge of a NACA0012 airfoil, chord length 0.53 m. One blue line represents one minute of ice accretion. Further boundary conditions (rime): $U_\infty = 58$ m/s, $t_{\text{acc}} = 8$ min; (glaze): $U_\infty = 102.8$ m/s, $t_{\text{acc}} = 7$ min. Data from computations with TAUICE by Jan Steiner.



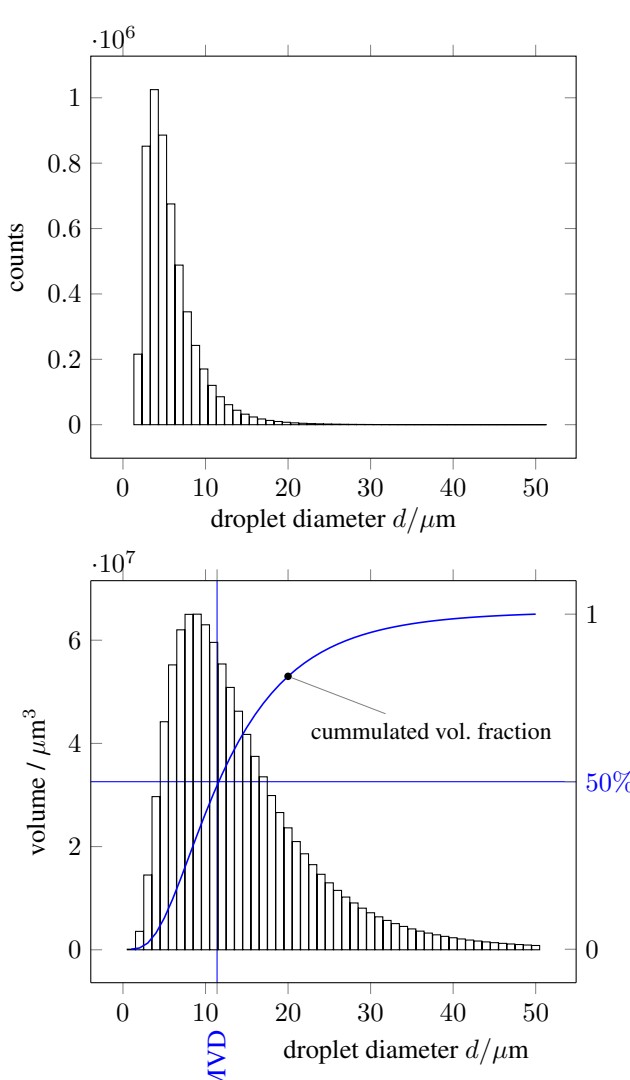

**Figure 2.** Determination of the median volume diameter (MVD).



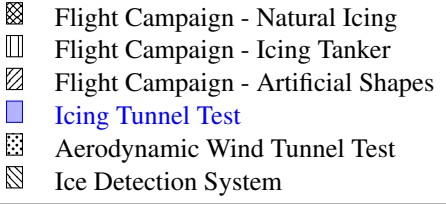

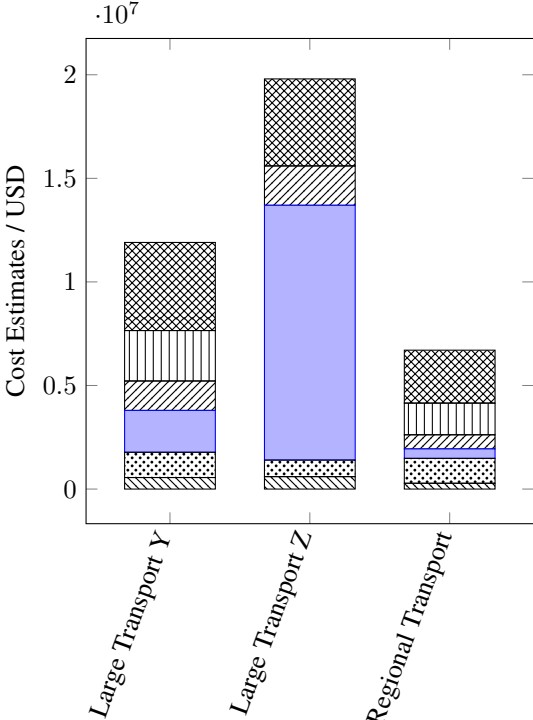

**Figure 3.** Cost estimation for aircraft icing certification according to FAR Part 25 Appendix O based on the working group report on supercooled large droplet rulemaking IPHWG (2005).





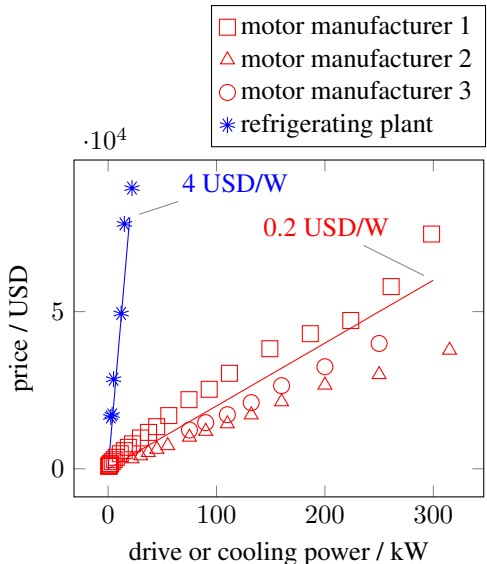

**Figure 4.** Pricing as of 2017 for asynchronous motors from different manufactures and for $CO_2$ refrigerating plants before tax and tolls.

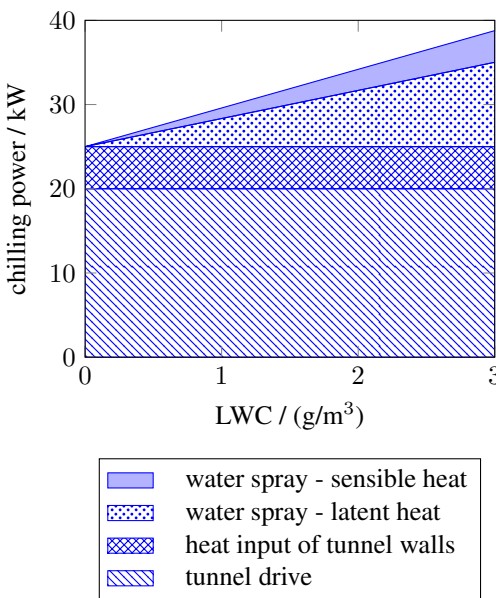

**Figure 5.** Required chilling power decomposition as a function of the liquid water content for typical operational conditions of the Braunschweig Icing Wind Tunnel.



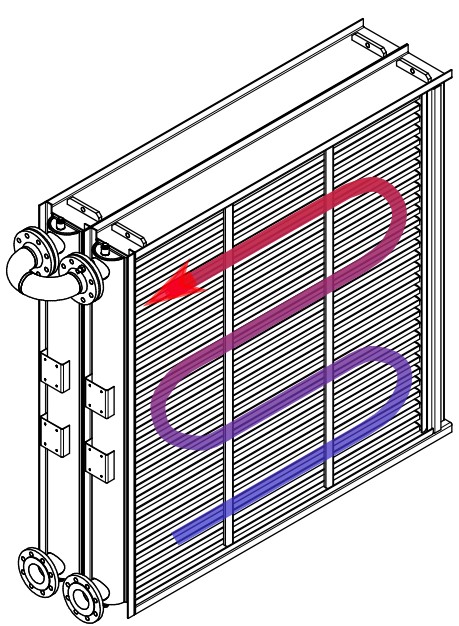

**Figure 6.** Heat exchanger inside the icing wind tunnel.

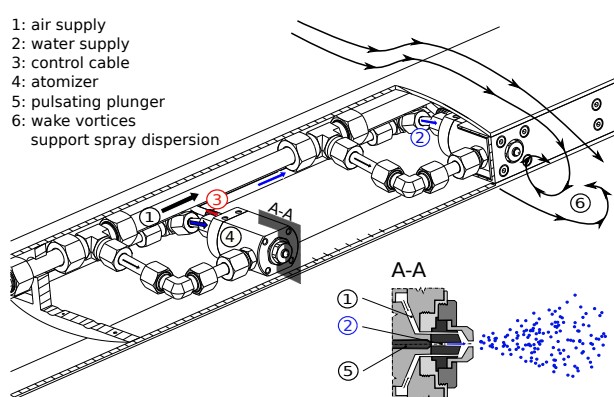

**Figure 7.** Spray system for low MVD and low LWC.



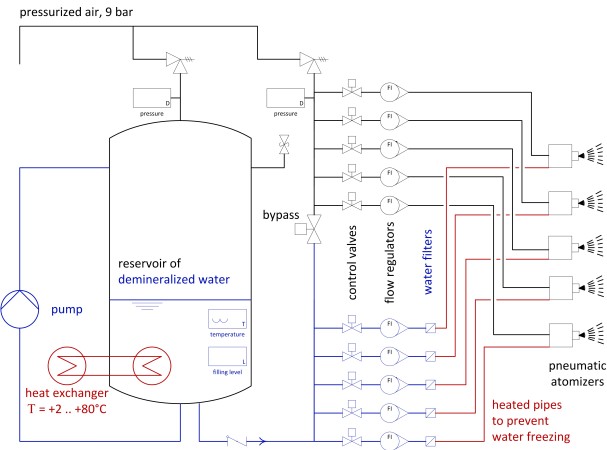

**Figure 8.** Spray conditioning system.

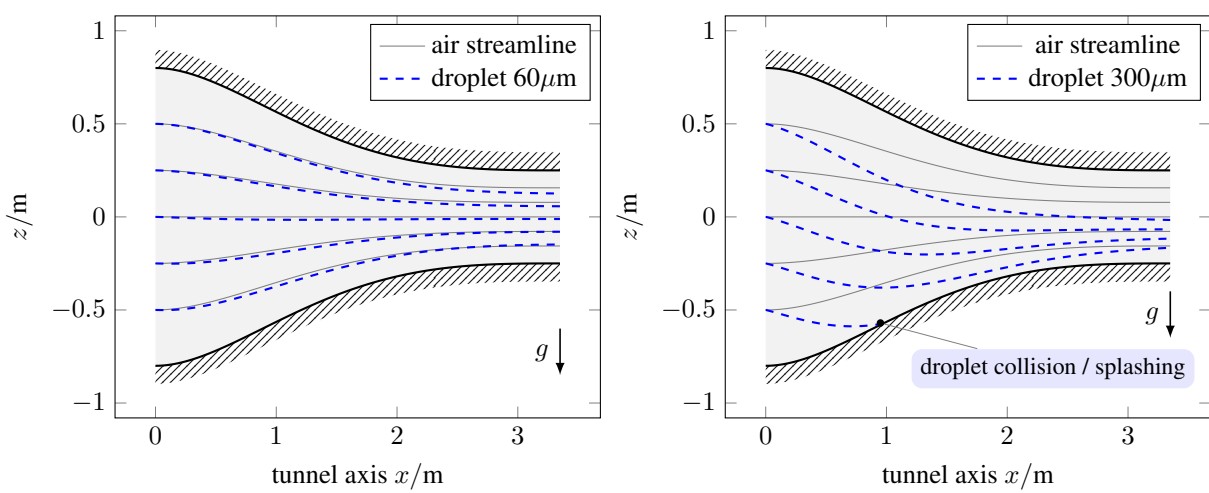

**Figure 9.** Droplet trajectories in the wind tunnel nozzle and their deviation from the air streamlines under the influence of gravity. Large droplets may collide with the tunnel walls thereby altering the droplet size distribution.





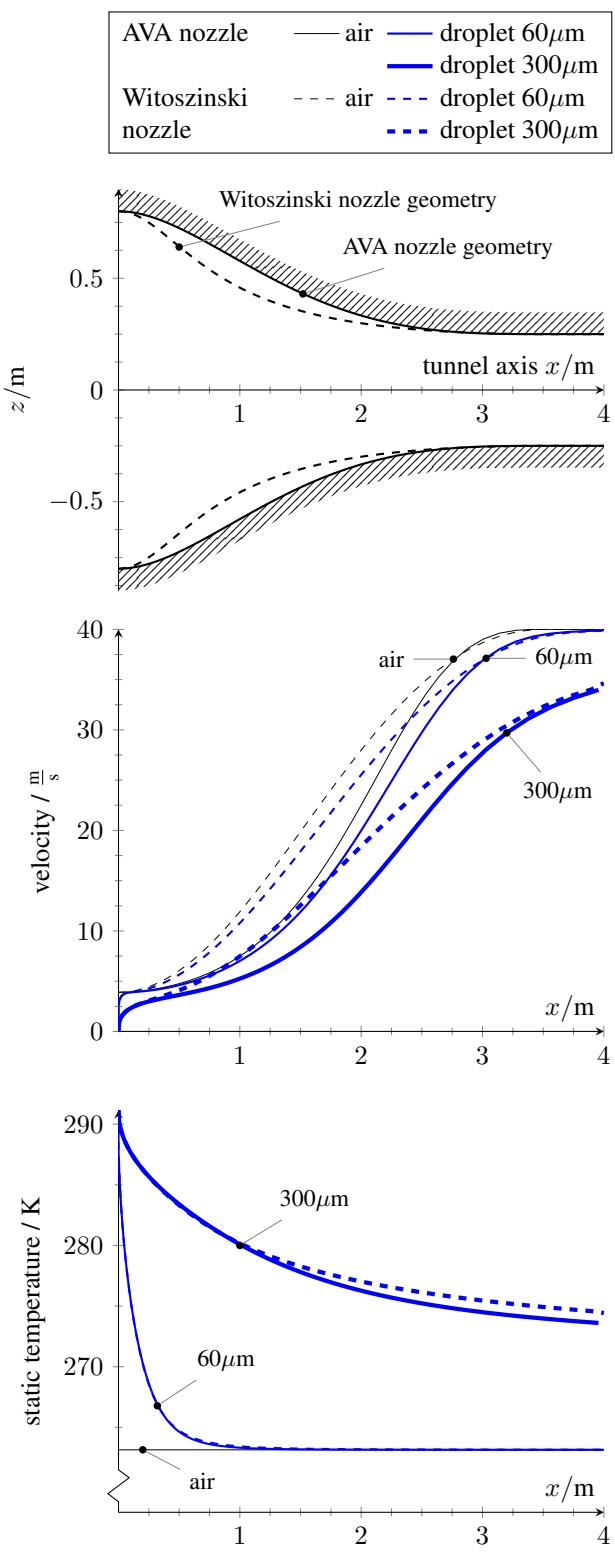

**Figure 10.** Velocity and temperature slope for air and two different sized droplets inside two tunnel nozzle configurations.




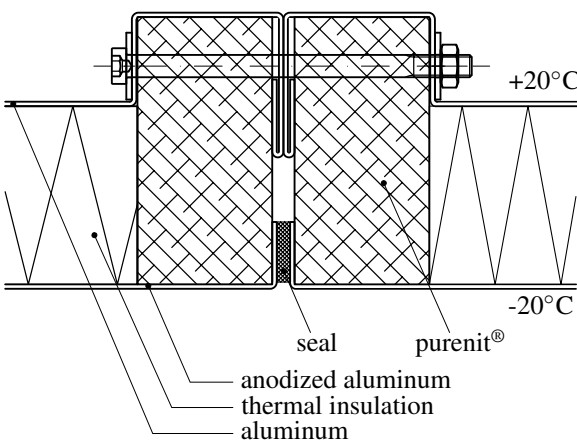

**Figure 11.** Mechanical design of the icing wind tunnel walls.

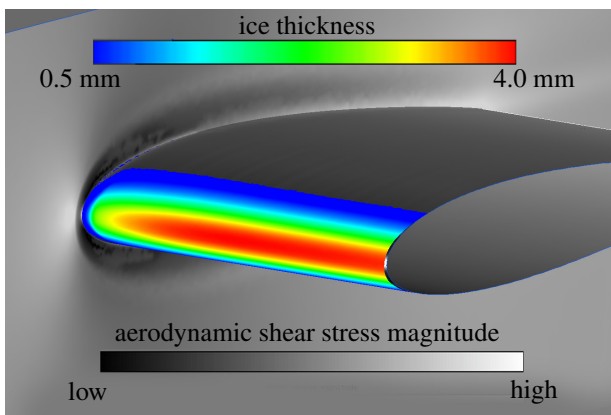

**Figure 12.** The junction flow between a NACA0012 airfoil (c = 75 cm, $\alpha = 0°$) and wind-tunnel side wall creates a horseshoe vortex, which is altering the ice accretion. $U_\infty = 40\frac{m}{s}$, $T_\infty = -10\,°C$, LWC=1$\frac{g}{m^3}$, IWC=0.





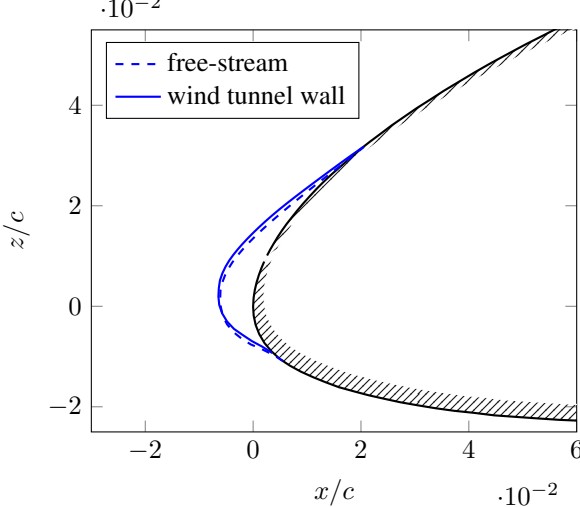

**Figure 13.** Inviscid wall effect on ice accretion for a AH-94-W-145 airfoil, c=0.75 m, $U_\infty$=40 m/s, $T_\infty = -10\ ^\circ C$, LWC=0.3 g/m$^3$, MVD=15 $\mu$m, IWC=0.

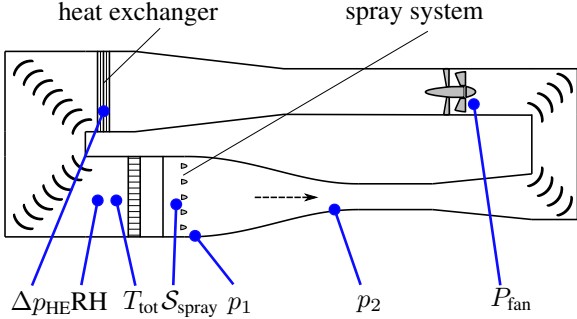

**Figure 14.** Basic instrumentation of the Braunschweig Icing Wind Tunnel.

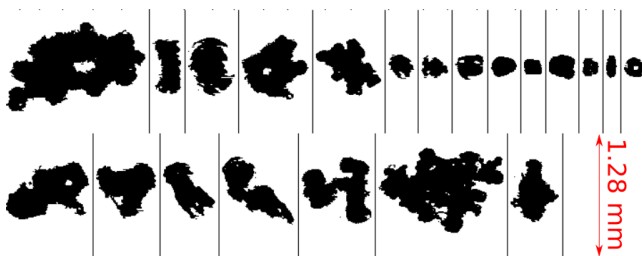

**Figure 15.** Ice particle images captured by 2DS probe in stratiform cloud region during the Darwin Campaign, $MMD \approx 150\mu m$, $IWC \approx 1.2g/m^3$, $T_\infty \approx -10\ ^\circ C$





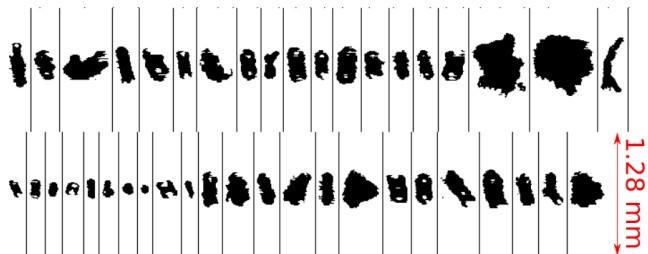

**Figure 16.** Ice particle images captured by 2DS probe in convective cloud region during the Darwin Campaign, $MMD \approx 80\mu m$, $IWC \approx 3.0 g/m^3$, $T_\infty \approx -10\,^\circ C$

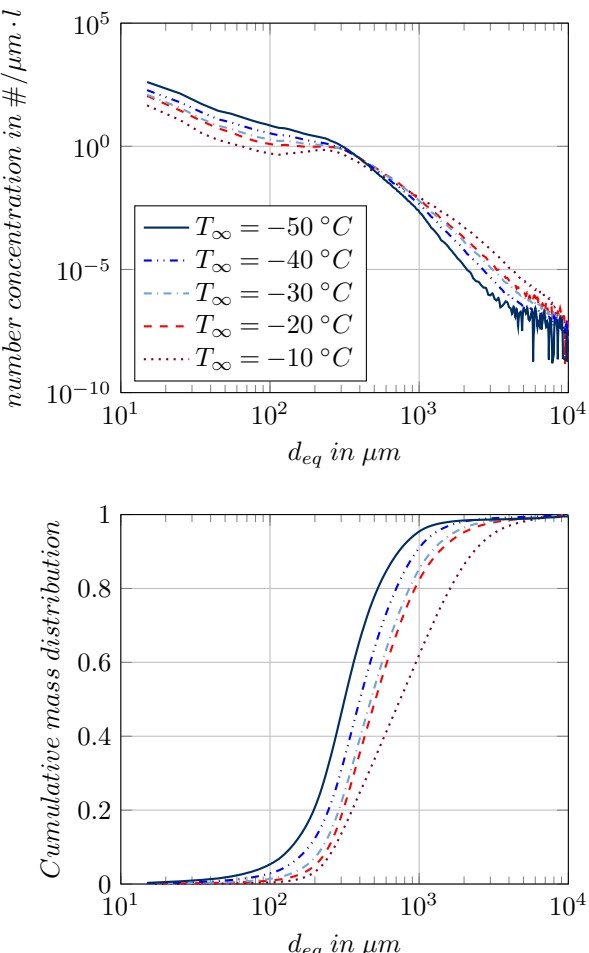

**Figure 17.** Ice particle size distribution (MMD) based on HAIC Falcon 20 flight measurements from Darwin and Cayenne, dependence on ambient air temperature



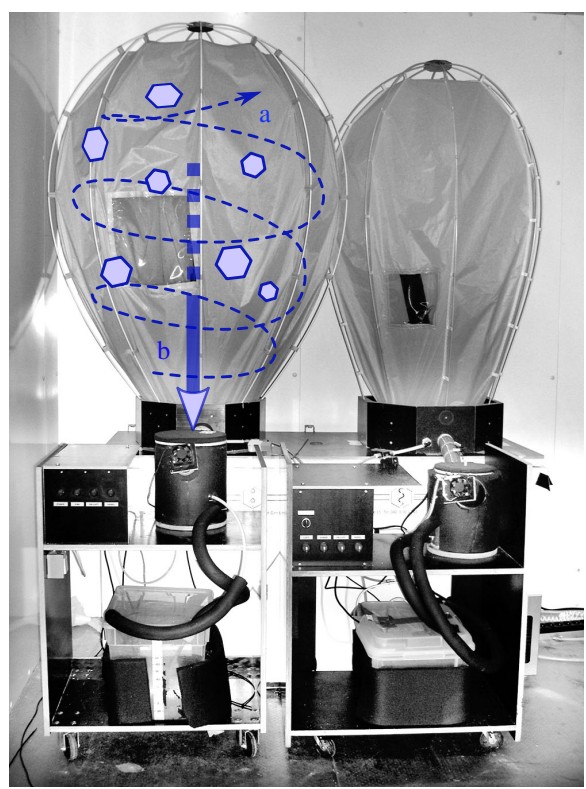

**Figure 18.** Ice crystal generators. A jet of air (a) circulates the flow allowing for ice crystal growth. When the crystals are large enough, they fall into a chest freezer (b).

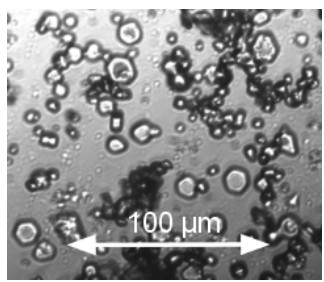

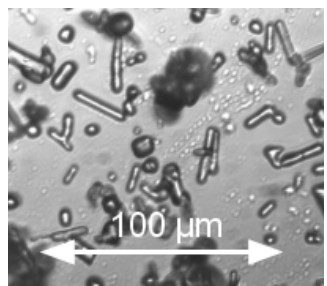

**Figure 19.** Microscopic images of ice particles inside the cloud chambers. *Left:* plate crystals. *Right:* Needles.



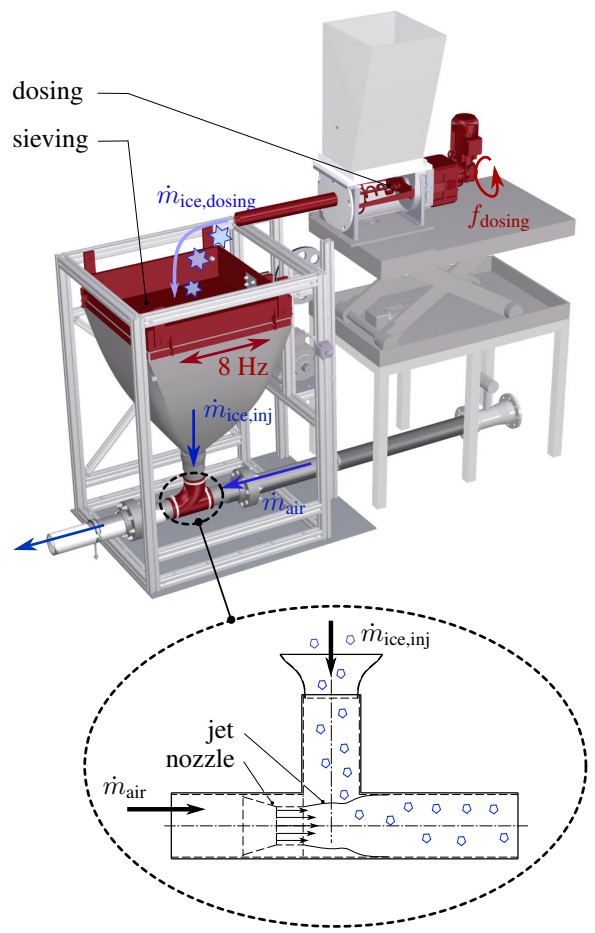

**Figure 20.** Assembly for ice particle dosing and fluidization.

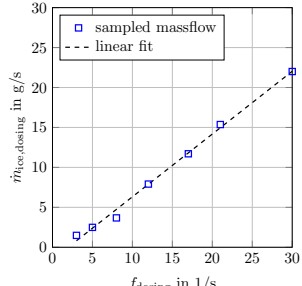
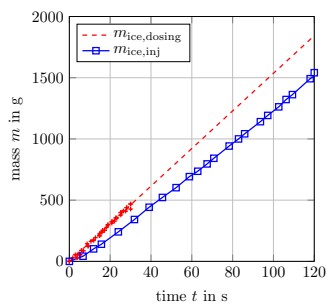

**Figure 21.** Ice particle massflow. *Left:* Dependence on dosing machine frequency. *Right:* Ice particle supply over time at a dosing frequency of $f_{dosing} = 21$ Hz.



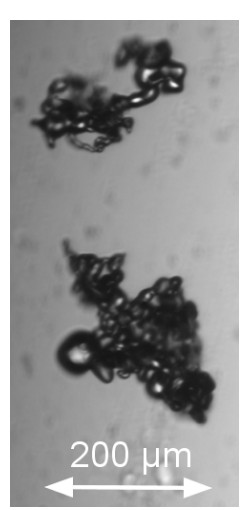

**Figure 22.** Ice particle structure after storage, dosing and sieving procedure

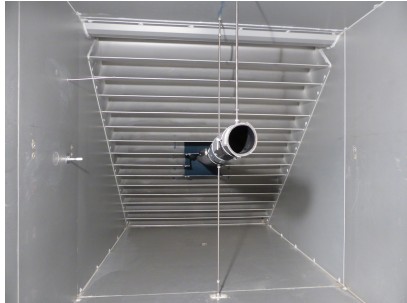

**Figure 23.** Ice particle conveyance pipe inside the settling chamber of the Braunschweig Icing Wind Tunnel





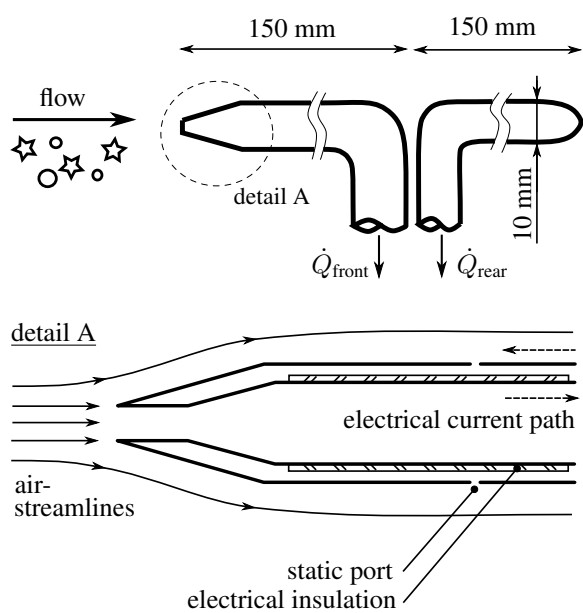

**Figure 24.** Isokinetic probe to measure the total water content TWC, developed at Cranfield university.

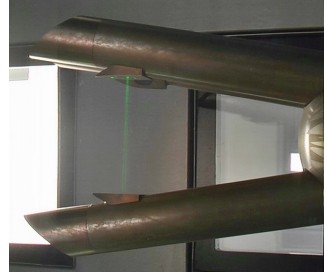
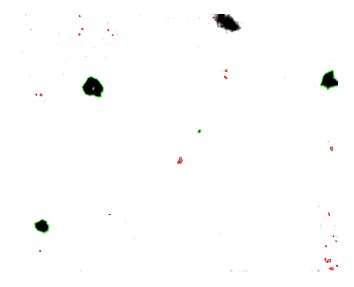

**Figure 25.** Ice particle detection by means of C-HSI probe. *Left:* Trigger beam induced by HSI. *Right:* Evaluation of shadow images.





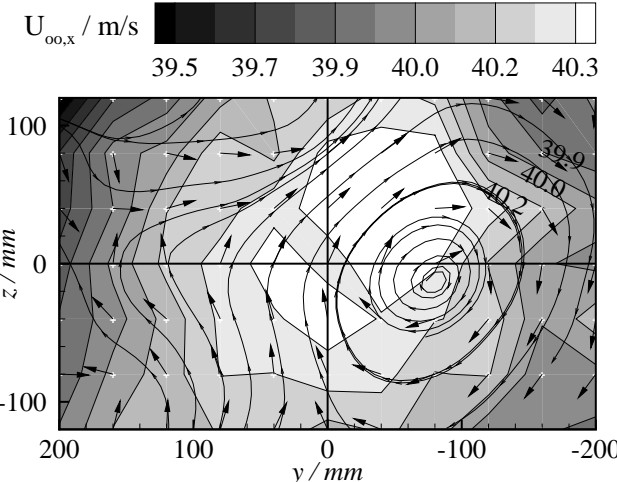

**Figure 26.** Spatial distribution of axial flow velocity $U_{\infty,x}$, for the ice-crystal-icing setup.

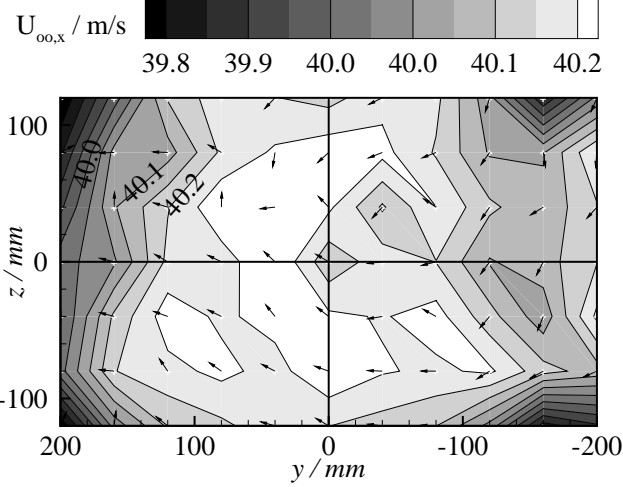

**Figure 27.** Spatial distribution of axial flow velocity $U_{\infty,x}$, for the standard setup.





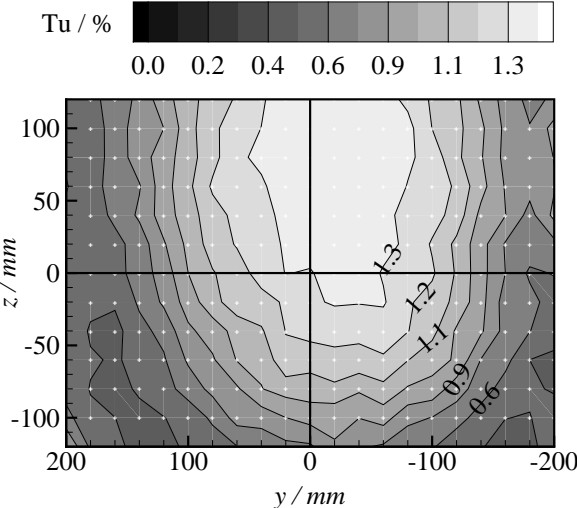

**Figure 28.** Spatial distribution of turbulence intensity for the mixed phase configuration

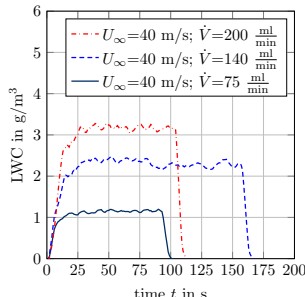

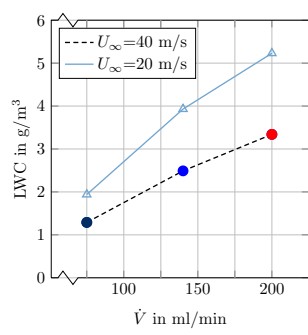

**Figure 29.** Liquid water content calibration based on IKP measurements. *Left:* LWC history. *Right:* LWC depending on supplied water flow-rate.

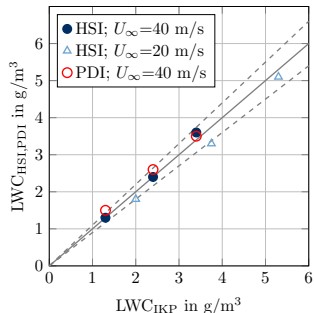

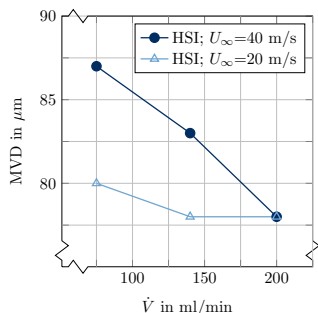

**Figure 30.** Comparison of HSI, PDI and IKP measurements for supercooled droplet conditions. *Left:* Optical array probes vs. IKP. *Right:* MVD vs. LWC.




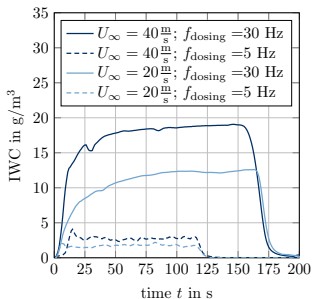
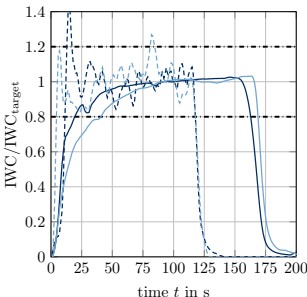

**Figure 31.** IWC time history based on IKP measurements. *Left:* IWC history. *Right:* Normalized IWC history.

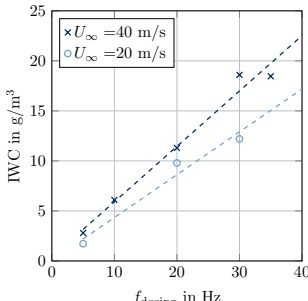
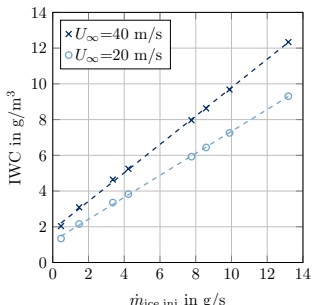

**Figure 32.** Ice water content depending on dosing frequency $f_{dosing}$ and injected ice particle flow $\dot{m}_{ice,inj}$

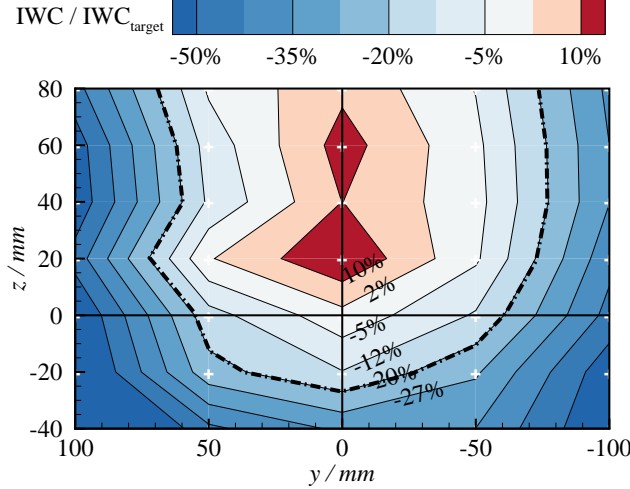

**Figure 33.** Percentaged deviation of IWC from centerline value, $U_\infty$=40 m/s, $f_{dosing}$=20 Hz.




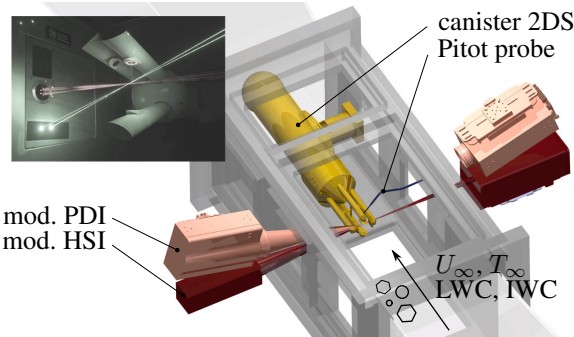

**Figure 34.** Setup of the probe inter-comparison exercise. Alternatively to the canister 2DS probe, which is shown here, the canister HSI probe was mounted. *Upper left:* C-HSI probe mounted inside the IWT, modular PDI and HSI laser beams are adjusted close to the C-HSI sampling volume

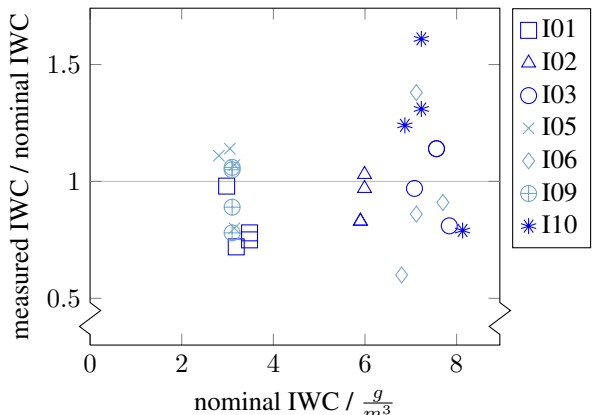

**Figure 35.** Results of the probe intercomparison between 2DS-LAMP, 2DS-DLR, M-HSI, C-HSI for seven different test points at glaciated icing conditions

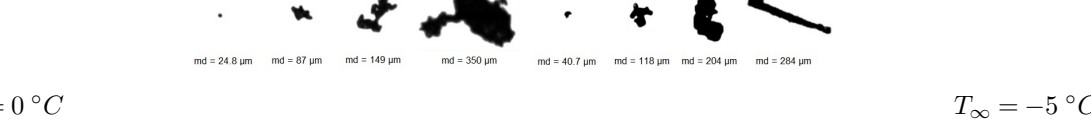

$T_\infty = 0\,^\circ C$ $T_\infty = -5\,^\circ C$

**Figure 36.** Ice particle images captured by the C-HSI probe inside the Braunschweig Icing Wind Tunnel test section

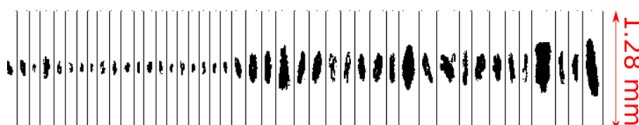

**Figure 37.** Ice particle images captured by the 2D-S probe inside the Braunschweig Icing Wind Tunnel test section, $MMD = 79\mu m$, $IWC = 3.2g/m^3$, $T_\infty = -15\,^\circ C$





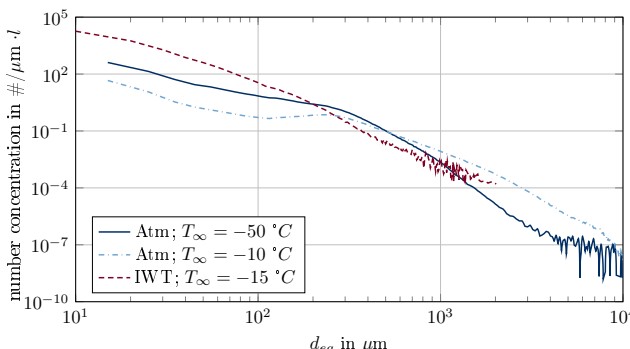

**Figure 38.** Particle size distribution: Comparison of artificial ice particle cloud to atmospheric cloud conditions

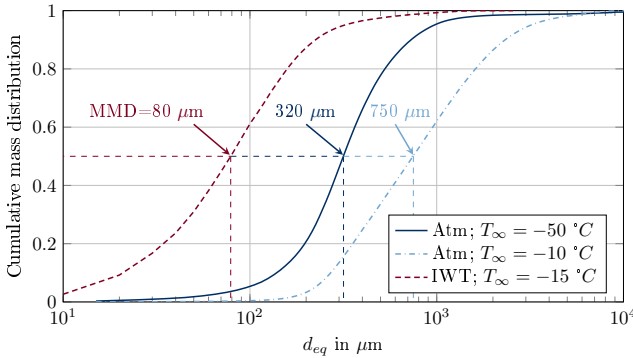

**Figure 39.** Cumulative particle size distribution: Comparison of artificial ice particle cloud to atmospheric cloud conditions

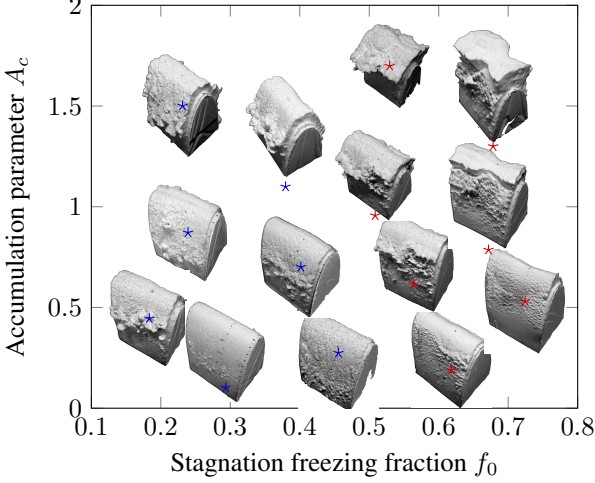

**Figure 40.** Experimentally observed SLD ice shapes on a NACA0012 leading edge in the parameter space of $f_0$ and $A_c$, Re=$2 \cdot 10^6$, MVD=$80\mu$m, AoA=$0°$. Results reproduced from Steiner and Bansmer (2016).




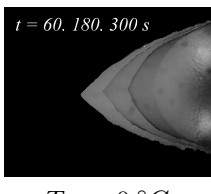

$T_\infty = 0\,°C$

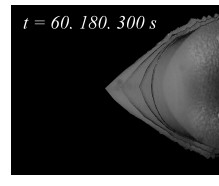

$T_\infty = -5\,°C$

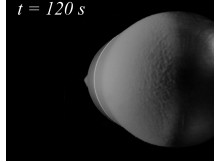

$T_\infty = -15\,°C$

**Figure 41.** Side view of ice accretion shapes on a cylinder for $TWC = 12g/m^3$ , $m_r = 0.12$ and $U_\infty = 40\ m/s$, with a variation of temperature

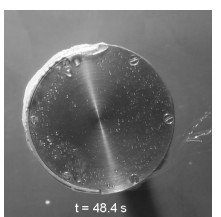

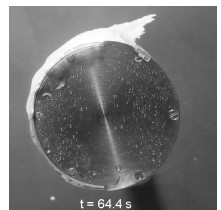

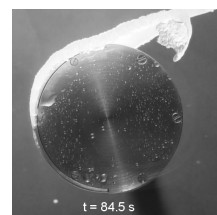

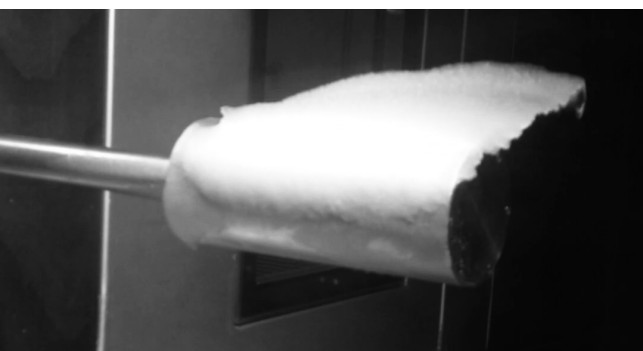

**Figure 42.** Side view of ice accretion when heating the cylinder, $T_\infty = -5\ °C, TWC = 6.8g/m^3$ and $m_r = 0.5$

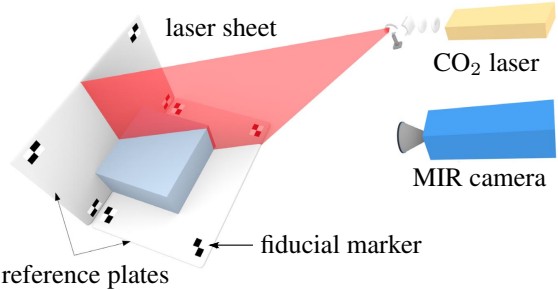

**Figure 43.** Infrared laser line scanning technique. The mid-infrared camera records the distorted $CO_2$-laser line on the ice surface. With a prior calibration, the 3D geometry of the object can be reconstructed. Image reproduced from Gong and Bansmer (2015a).



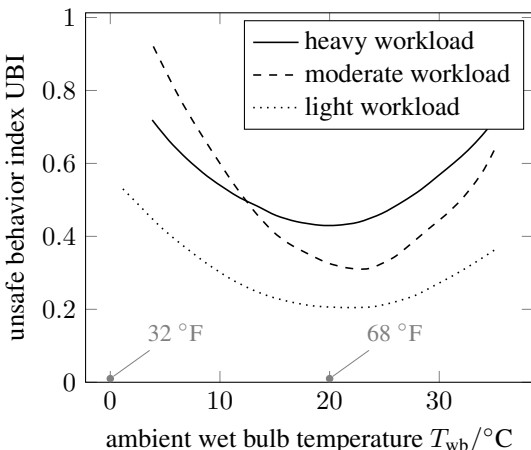

**Figure 44.** Even for appropriate clothing, exposure to cold temperatures yields an increase in unsafe behavior. Data reproduced from Ramsey et al. (1983).

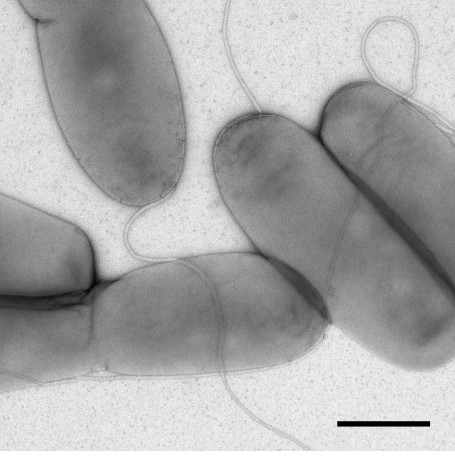

**Figure 45.** *Legionella pneumophila*, strain Corby. Transmission electron microscopy, negative contrast, scale marker = 500 nm. Picture by Norbert Bannert, Gudrun Holland / Robert Koch Institute.







**explanations**

① Braunschweig Icing Wind Tunnel
② control room
③ ice crystal / water droplet injection
④ tunnel refrigeration system
⑤ reservoir of cooling agent
⑥ control valves
⑦ surge tank filled with nitrogen
⑧ ice crystal generation and conveyance system (IGS)
⑨ cloud chamber for the production of natural ice crystals
⑩ $CO_2$ refrigeration unit for the IGS
⑪ radial fan for pneumatic conveyance
⑫ aftercooler
⑬ ice particle dosing device
⑭ plenum for ice particle distribution

**Figure 46.** Overview of the Braunschweig Icing Wind Tunnel and its adjacent components.