# Peer review of "Design, Construction and Commissioning of the Braunschweig Icing Wind Tunnel"

_Atmospheric Measurement Techniques, 2017_

## Referee Comment (RC1) · D W Hammond (Referee) · 13 Dec 2017

The paper is of very great interest to the users and makers of icing test facilities and a good many others. The paper includes fresh information which is specifically about the exciting new facility but also contains much text about the generalities of icing. It is my firm opinion that this general material is dealt with much better elsewhere and should be edited out of the paper.

Abstract, slightly wordy. The main message is relegated to lines 8 & 9 and dealt with almost in passing. I suggest something along the lines of "The unique aspect of this facility is ...(the combination of an icing tunnel with a cloud chamber system for making ice particles which are more realistic than those usually used for mixed phase and ice

crystal icing experiments).

Introduction. Paragraph 1 & 2 OK. Paragraph 3 might be stronger if the influence of temperature & heat capacity (water & ice) are mentioned alongside latent heat.

Page 2, line 13 to page 4, line 31 needs a good shake up to omit the generalities and concentrate on the aspects of direct significant to the development of an icing tunnel, this particular one.

Design & Construction....(starts Page 5) Lines 6-10. The rational is incompletely described. No mention of air speed of opportunities taken of lost by opting for a smaller or larger section.

Page 8 Lines 10 to 13. The LWC range for lager droplet mode is given but it is not given for the smaller droplet mode.

Page 11 lines 15 to 20. It would be nice to say a bit about how well this construction method worked out in practice. Was it easy to control leaks? How easy was it to avoid steps in the wall?

Page 11 lines 21 to Page 12, line 2. I find this rather unclear. Specifically, the figure (12) appears to show ice thickness. It drops of at the ends of the aerofoil but this could be for a number of reasons not discussed in the text. We are left wondering about the conditions modeled and learn nothing of the predicted velocity fiend which I would surely prove the central point more effectively.

Page 12 lines 11. Here the heat exchanger is said to be off galvanised mild steel. On page 7, line 28, we are told that it is of a mix of steels, galvanised together and coated with an epoxy based film. So which is it? There is also a reference to sub sub sub section in some other document which I recon should not be in the text.

Ice Crystals & mixed phase... (starts page 13) Page 14, lines 17 to 26. To me this is a bit muddled. For me the problem is exemplified best by the description of the blobs in figures 16 as "crystals" rather than hydrometers or lumps. They are aggregates,

possibly of some single crystal particles but also, I would suggest, some degree of rime growth. Other references to "ice crystals" through the rest of the text need to be checked to ensure that the vocabulary does justice to the work.

Please also note the supplement to this comment:
https://www.atmos-meas-tech-discuss.net/amt-2017-356/amt-2017-356-RC1-supplement.pdf
* * *

---

## Referee Comment (RC2) · D. Fuleki (Referee) · 5 Jan 2018

[referee-annotated manuscript omitted]

---

## Author Comment (AC1) · 2 Feb 2018

**Response to the review of David Hammond.**

On behalf of my coauthors, I am very thankful for your review on the paper "Design, Construction and Commissioning of the Braunschweig Icing Wind Tunnel". Please find our comments below, which will be implemented in our paper accordingly.

The paper is of very great interest to the users and makers of icing test facilities and a good many others. The paper includes fresh information which is specifically about the exciting new facility but also contains much text about the generalities of icing. It is my firm opinion that this general material is dealt with much better elsewhere and should be edited out of the paper.

Writing that introduction & manuscript was a balancing act between two perspectives. On the one hand, we wanted to present some new ideas to experienced icing researchers. On the other hand, we wanted to explain our experiences to a general audience, to someone who is new in the topic, and who does not know all the specific vocabulary. It is therefore our feeling, that some of the general material provided shall not simply be omitted. Nevertheless, we agree with you, that this manuscript shall not become a textbook, likewise. In consequence, we will modify the introduction, see also our later comments.

Abstract, slightly wordy. The main message is relegated to lines 8 & 9 and dealt with almost in passing. I suggest something along the lines of "The unique aspect of this facility is ...(the combination of an icing tunnel with a cloud chamber system for making ice particles which are more realistic than those usually used for mixed phase and ice crystal icing experiments).

We will point out the unique aspect of our tunnel more specifically in the abstract. Your suggested sentence is appreciated.

Introduction. Paragraph 1 & 2 OK. Paragraph 3 might be stronger if the influence of temperature & heat capacity (water & ice) are mentioned alongside latent heat.

We are not sure on how to answer that question. The nucleation process itself is not primarily influenced by the heat capacity. Is it the reviewer's intention to point out the anomalies of water? We kindly ask the reviewer to clarify on that point. Of course we will implement his suggestions afterwards.

**Page 2, line 13 to page 4, line 31 needs a good shake up to omit the generalities and concentrate on the aspects of direct significant to the development of an icing tunnel, this particular one.**

Starting at line 13, the first paragraph summarizes the important boundary conditions of an icing wind tunnel experiment. It is our intention to address a broad readership with this article. Not every reader is familiar with either the aerodynamic parameters or the icing parameters. An explanation of these parameters seems only logical.

The next paragraph was further elucidating on the MVD. We have combinded this paragraph with the enumeration of boundary conditions.

We will further omit the paragraph, which explains how the MVD is determined with a given droplet size spectrum.

Page 3 Line 30 then elucidates on quantitative values for boundary conditions. The second reviewer even suggested adding some quantitative information on ice crystals at that

occasion. From our point of view, it is very important to provide that kind of quantitative information for a general audience.

We will then reduce the paragraph on scaling laws significantly, but we will mention a reference to them.

The last two paragraphs explain are the outline for the following sections, and summarize the introduction. We suggest to keep them.

Design & Construction....(starts Page 5) Lines 6-10. The rational is incompletely described. No mention of air speed of opportunities taken of lost by opting for a smaller or larger section.

Based on the Reynolds number ( $\sim$ V\*I) criterion, it is indeed a compromise between velocity (V) and test section dimension (I) that has to be made. The consideration of wind tunnel and chilling power is even more significant, which we have explained in the subsequent paragraph, because it directly relates to your investment costs. From a pure mathematical point, the two parameters of drive Power ( $\sim$ V3 \* I2) and Reynolds number will uniquely determine the velocity V and the size I.

From an engineering perspective, there is of course some choice between test section size and velocity. From our point of view, the test section size of 0.5 m x 0.5 m was quite important. Below that size, it would not have been possible to install the canister probes (shown in Figure 34) or any other significant aircraft subsystem. A test section, which is even larger would have reduced the maximum speed in the test section at a given fan power. We did not want to go below 40 m/s because our major customers indicated a desire for even higher speeds. Finally, the tunnel dimensions are also a constraint of available space.

We will elaborate more in detail in our manuscript on our design choices.

Page 8 Lines 10 to 13. The LWC range for lager droplet mode is given but it is not given for the smaller droplet mode.

We have added that information in the manuscript.

Page 11 lines 15 to 20. It would be nice to say a bit about how well this construction method worked out in practice. Was it easy to control leaks? How easy was it to avoid steps in the wall?

A company that is specialized on sheet metal forming manufactured the parts. Small tolerances of all parts were ascertained. The step size between the segments is negligible. Since the flanges were fastened with many screws, the leakage was minimal. To make the joints fully watertight, we sealed the inner groove with another acrylic compound.

We will add some statements about these practical aspects.

Page 11 lines 21 to Page 12, line 2. I find this rather unclear. Specifically, the figure (12) appears to show ice thickness. It drops of at the ends of the aerofoil but this could be for a number of reasons not discussed in the text. We are left wondering about the conditions modeled and learn nothing of the predicted velocity fiend which I would surely prove the central point more effectively.

The flow around a NACA0012 airfoil is simulated. It is a 3D RANS computation that predicts the junction flow between airfoil and sidewall. This junction flow creates a horseshoe vortex.

We created an additional picture, see below, to visualize this horseshoe vortex. The velocity in spanwise direction is shown there for different slices. The gray contour represents the shear stress at the sidewall. We simulated LWC 1g/m3 and MVD 20µm at 40m/s, T=-10°C and Re= $2^{*10^6}$ . Note that the boundary conditions are already mentioned in the figure caption. Indeed, the horseshoe vortex alters the velocity distribution at the junction. When velocity is altered, the heat transfer is influenced and so the ice accretion rate (glaze ice). The flow solver is coupled to a trajectory solver and an ice accretion model, see the references already provided. The ice thickness shown in Figure 12 results after an ice accretion time of 120 seconds. We will include this additional information in the manuscript.

Page 12 lines 11. Here the heat exchanger is said to be off galvanised mild steel. On page 7, line 28, we are told that it is of a mix of steels, galvanised together and coated with an epoxy based film. So which is it? There is also a reference to sub sub sub section in some other document which I recon should not be in the text.

Indeed, the statements of page 7 are correct. The intention of page 12 is to elucidate on the corrosion process, if the epoxy coating were not be present / or deteriorated. We will clarify this point and correct the reference.

Ice Crystals & mixed phase... (starts page 13) Page 14, lines 17 to 26. To me this is a bit muddled. For me the problem is exemplified best by the description of the blobs in figures 16 as "crystals" rather than hydrometers or lumps. They are aggregates, possibly of some single crystal particles but also, I would suggest, some degree of rime growth. Other references to "ice crystals" through the rest of the text need to be checked to ensure that the vocabulary does justice to the work.

We agree with the reviewer that the wording ice crystals may be too specific for the various types of ice particles found in convective clouds. The definition of ice crystal is given in the AMS glossary: http://glossary.ametsoc.org/wiki/Ice\_crystals

For simplicity we have chosen to use "ice particles" instead and only "ice crystals" for the column and capped column type ones.

---

## Editor Decision (ED1)

Dear Stephan Bansmer and Co-authors,

thanks for your detailed responses and the revised manuscript which is greatly improved.

There are some further points that need revision before this manuscript can be published, see below:

**General comments:**

Your manuscript contains an overwhelming number of figures. Several of them could be combined into fewer figures (details below), other figures might not be as relevant that they would need to appear in the main manuscript. Please consider to move some of your figures into a supplement.

Formatting of units: (citing from the manuscript preparation guidelines) "Also units in the denominator should be formatted with negative exponents (e.g. km h-1 instead of km/h)."

(Page and line numbers refer to manuscript version 4 in your MS records, i.e. the version prior to the nomenclature version.)

Specific comments:

Abstract:

p1, I5: "...static temperature..." = static air temperature?

p1, I8: "... than usually used..." - what is usually used?

**p1, I8/9:** "Ice water contents up to 20 g=m3 can be simulated." "Simulated" sounds like you talk about numerical model simulation, maybe better "generated"?

**p1, I9:** "... measurement techniques for particle sizing have performed..." Either "have been performed" or "are performed".

**p1, l11:** "Finally, some applications of the icing wind tunnel are mentioned." Maybe replace "mentioned" with "presented" or "introduced".

**Introduction:**

**p2, I30:** You introduce "static air temperature" but then talk about "static temperature". Please be consistent throughout the manuscript!

**p3, I5:** "...for which the model..." Which model? It is the first time a model is mentioned, so you should explain more.

**p3**, **I7**: "super-cooled" – throughout the manuscript you use "supercooled" as well as "super-cooled", be consistent!

**p3, I30:** "European Aviation Safety Agency (EASA)" – already introduced in line 16, so use acronym only.

Section 2:

p4, l27: duplicate "was"

**p4, l28/29:** Better: "The space available at the installation site suddenly becomes a further design constraint."

p5, l8: "temperature bound" = "temperature threshold"?

**p5**, **l11**: "Figure 46" – figures should be numbered in the order they appear in the text!

p7, I9: "PLC" – this acronym has not been introduced!

p7, l16: figure13 – again, figures should be numbered in the order they appear in the text!

p7, l24: Check format of units.

p7, l32: "introduced" - better "installed"

p8, l1: "where as" - "whereas"

**p11, l19:** "... sealed with another acrylic compound." – specify "another" or use "an" instead of "another"

p11, l20: "..., no steps..." - do you mean "gaps"?

p12, l9: Is there a reference for TAUICE?

p12, l13: duplicate "of"

Section 3

p13, Section 2.9: Can you specify accuracies for the static pressure probes and the Vaisala HMT-337?

**p14, I32/Figure 16:** In the introduction you introduce Median Volume/Mass Diameters as important parameters, why do you use deq here?

**p14**, **l31** – **p15**, **l14**: I wonder whether this paragraph could be shortened. What value does it add to the manuscript?

**p16**, **I5**: "IWT" – the acronym has not been introduced yet, please write out.

**p16**, **I32/33**: "ice water content IWC" – IWC had been introduced in the introduction, no need to write out here.

Section 4:

p17, l15: "icing-wind-tunnel" – be consistent, either "icing-wind-tunnel" or "icing wind tunnel"

p17, l19: PDI – what does this acronym stand for?

p18, l1: typo: "user?s preference."

p18, l12-32: Is there no reference for the IKP?

**p18, l17:** "water vapour" – be consistent american english (or british english) throughout the manuscript.

**p18**, **l31/32**: Why do you turn off the shattering artefact removal feature of the software for the wind tunnel measurements? Don't you expect particle shattering to happen here, too? Especially in high IWC or LWC conditions! Furthermore, it becomes hard then to compare in situ field measurements and wind tunnel measurement when this treatment differs!

**p19, l15/16, l24:** "2DS" vs. "2D-S" – be consistent.

**p23, 17:** "...higher concentrations of small particles..." – could this not be due to particle shattering introducing numerous small particles (as the removal treatment was turned off)?

Section 5

**p23**, **l27**: "European Aviation Safety Agency" = EASA (acronym already introduced)

p24, l17: "generic test articles" – do you mean "test objects" or "test models"?

**p25, l10/l22:** "3-D" vs. "3D" – be consistent!

Acknowledgements:

p26, l12: typo "contact" - contract

Figures:

As mentioned before, the figures should be numbered in the order they are mentioned in the text.

Furthermore, I find that you have a huge number of figures. I suggest you have a good revision of which figures are important and less important figures could be moved to a supplement. Also, maybe some figures could be grouped together, or even eliminated. See following suggestions:

- The manuscript contains a number of technical figures, these could potentially be grouped together (and then labeled as subfigures), e.g. Figures 6 and 7; Figures concerning the ice particle generation and dosing, including the ice particle conveyance pipe; maybe you have further ideas where figures would fit together thematically. I also feel that some of these figures might be well suited for a supplement.
- Figures that could be joined together and labeled as subfigure (a) and (b): 14 + 15, 21 + 19, 25 + 26, 27 + 28, 34 + 35, 36 + 37, 38 + 39, 41 + 42
- Figure 45 is not needed and can be removed what value does it add to the manuscript?
- Figure 23 + 24: You show images for the IKP and HSI probes, but not for PDI and 2DS probes. Are these figures needed, what value do they add to the manuscript?

Figure 41: Where exactly is the cylinder? (It is not as obvious as in Figure 42.) Why are the side views of figures 41 and 42 so different, is Figure 41 a model simulation? Also, I think these two figures could be joined into one.

Units in figures: "Also units in the denominator should be formatted with negative exponents (e.g.  $\text{km h}^{-1}$  instead of km/h)."

---

## Author Response (AR2)

Dear Stephan Bansmer and Co-authors,

thanks for your detailed responses and the revised manuscript which is greatly improved.

There are some further points that need revision before this manuscript can be published, see below:
[Figure]

General comments:

Your manuscript contains an overwhelming number of figures. Several of them could be combined into fewer figures (details below), other figures might not be as relevant that they would need to appear in the main manuscript. Please consider to move some of your figures into a supplement.

Formatting of units: (citing from the manuscript preparation guidelines) "Also units in the denominator should be formatted with negative exponents (e.g. km h-1 instead of km/h)."

(Page and line numbers refer to manuscript version 4 in your MS records, i.e. the version prior to the nomenclature version.)

Specific comments:

*Abstract:*

**p1, l5:** "...static temperature..." = static air temperature?

**p1, l8:** "... than usually used..." – what is usually used?

**p1, l8/9:** "Ice water contents up to 20 g=m3 can be simulated." "Simulated" sounds like you talk about numerical model simulation, maybe better "generated"?

**p1, l9:** "... measurement techniques for particle sizing have performed..." Either "have been performed" or "are performed".

**p1, l11:** "Finally, some applications of the icing wind tunnel are mentioned." Maybe replace "mentioned" with "presented" or "introduced".

*Introduction:*

**p2, l30:** You introduce "static air temperature" but then talk about "static temperature". Please be consistent throughout the manuscript!

**p3, l5:** "...for which the model..." Which model? It is the first time a model is mentioned, so you should explain more.

**p3, l7:** "super-cooled" – throughout the manuscript you use "supercooled" as well as "super-cooled", be consistent!

**p3, l30:** "European Aviation Safety Agency (EASA)" – already introduced in line 16, so use acronym only.

*Section 2:*

**p4, l27:** duplicate "was"

**p4, l28/29:** Better: "The space available at the installation site suddenly becomes a further design constraint."

**p5, l8:** "temperature bound" = "temperature threshold"?

**p5, l11:** "Figure 46" – figures should be numbered in the order they appear in the text!

**p7, l9:** "PLC" – this acronym has not been introduced!

**p7, l16:** figure13 – again, figures should be numbered in the order they appear in the text!

**p7, l24:** Check format of units.

**p7, l32:** "introduced" – better "installed"

**p8, l1:** "where as" – "whereas"

**p11, l19:** "... sealed with another acrylic compound." – specify "another" or use "an" instead of "another"

**p11, l20:** "..., no steps..." – do you mean "gaps"?

**p12, l9:** Is there a reference for TAUICE?

**p12, l13:** duplicate "of"

*Section 3*

**p13, Section 2.9:** Can you specify accuracies for the static pressure probes and the Vaisala HMT-337?

**p14, l32/Figure 16:** In the introduction you introduce Median Volume/Mass Diameters as important parameters, why do you use $d_{eq}$ here?

**p14, l31 – p15, l14:** I wonder whether this paragraph could be shortened. What value does it add to the manuscript?

**p16, l5:** "IWT" – the acronym has not been introduced yet, please write out

**p16, l32/33:** "ice water content IWC" – IWC had been introduced in the introduction, no need to write out here.

*Section 4:*

**p17, l15:** "icing-wind-tunnel" – be consistent, either "icing-wind-tunnel" or "icing wind tunnel"

**p17, l19:** PDI – what does this acronym stand for?

**p18, l1:** typo: "user?s preference."

**p18, l12-32:** Is there no reference for the IKP?

**p18, l17:** "water vapour" – be consistent american english (or british english) throughout the manuscript.

**p18, l31/32:** Why do you turn off the shattering artefact removal feature of the software for the wind tunnel measurements? Don't you expect particle shattering to happen here, too? Especially in high IWC or LWC conditions! Furthermore, it becomes hard then to compare in situ field measurements and wind tunnel measurement when this treatment differs!

**p19, l15/16, l24:** "2DS" vs. "2D-S" – be consistent.

**p23, l7:** "…higher concentrations of small particles…" – could this not be due to particle shattering introducing numerous small particles (as the removal treatment was turned off)?

*Section 5*

**p23, l27:** "European Aviation Safety Agency" = EASA (acronym already introduced)

**p24, l17:** "generic test articles" – do you mean "test objects" or "test models"?

**p25, l10/l22:** "3-D" vs. "3D" – be consistent!

*Acknowledgements:*

**p26, l12:** typo "contact" – contract

*Figures:*

As mentioned before, the figures should be numbered in the order they are mentioned in the text.

Furthermore, I find that you have a huge number of figures. I suggest you have a good revision of which figures are important and less important figures could be moved to a supplement. Also, maybe some figures could be grouped together, or even eliminated. See following suggestions:

- The manuscript contains a number of technical figures, these could potentially be grouped together (and then labeled as subfigures) , e.g. Figures 6 and 7; Figures concerning the ice particle generation and dosing, including the ice particle conveyance pipe; maybe you have further ideas where figures would fit together thematically. I also feel that some of these figures might be well suited for a supplement.
- Figures that could be joined together and labeled as subfigure (a) and (b): 14 + 15, 21 + 19, 25 + 26, 27 + 28, 34 + 35, 36 + 37, 38 + 39, 41 + 42
- Figure 45 is not needed and can be removed – what value does it add to the manuscript?
- Figure 23 + 24: You show images for the IKP and HSI probes, but not for PDI and 2DS probes. Are these figures needed, what value do they add to the manuscript?

Figure 41: Where exactly is the cylinder? (It is not as obvious as in Figure 42.) Why are the side views of figures 41 and 42 so different, is Figure 41 a model simulation? Also, I think these two figures could be joined into one.

Units in figures: "Also units in the denominator should be formatted with negative exponents (e.g. km h$^{-1}$ instead of km/h)."

[revised manuscript text omitted]

---

## Editor Decision (ED2)

Dear Stephan Bansmer,

thanks for the replies and revision of the manuscript. I do have some follow up questions and further comments, please see below.

Follow up questions – I repeat my original question and your response (typeset in bold) in order to keep the context before asking my follow up question (typeset in italic):

p1, l8: "... than usually used..." – what is usually used?

**We stayed with the original wording, which came from the reviewer (British native speaker). Is that okay with you?**

I did not say that the wording needs changing, but it needs explanation! You are talking about facilities ", usually used" but it is not clear what facilities are usually used, name it!

p1, l9: "... measurement techniques for particle sizing have performed..." Either "have been performed" or "are performed".

We stayed with the original wording, which came from the reviewer (Canadian native speaker). The meaning is "The techniques have performed well." Is that okay with you?

The reviewer's suggestion was "... for particle sizing are performed on ice crystals" (which was one of my alternatives, too). If that does not reflect your meaning of the sentence it might be better to rephrase the sentence!

p5, l11: "Figure 46" – figures should be numbered in the order they appear in the text!

We have put the large image on the facility deliberately as the last picture in the manuscript. I guess it depends how the final paper is typesetted... That's why the figure numbering is a bit odd at this place. We can move the figure of course, if you insist...

As I said, figures should be numbered in the order they appear. If you want the figure to appear at an exposed position, you could, I assume, easily move it to number 1 by referring to it early in your manuscript.

p7, l16: figure13 – again, figures should be numbered in the order they appear in the text!

Here it is more difficult. The Figure 13 is explained properly much later in the manuscript (section 2.9). It was just one reviewer, who wanted to have information on where the heat exchanger is placed inside the tunnel's circuit. That's why we created the link here. What do you suggest?

As I said, figures should be numbered in the order they appear. You could for example add a reference to the section where the figure is described in more detail here!

p12, l9: Is there a reference for TAUICE?

No

Is it an abbreviation for something then? I feel there should be a few words about the code to give the reader an idea what it is used for etc.

p14, l32/Figure 16: In the introduction you introduce Median Volume/Mass Diameters as important parameters, why do you use d eq here?

The area equivalent diameter Deq defined as the diameter of a circle of the same area as the shaded pixelnumber for each particle is one way to deal with size information of particles. The information on particle shape is lost for Deq but hasn't been the focus of this study. In general, comparibility to the airborne measurements was aimed at and therefore Deq has been used.

We added this information in the manuscript.

Maybe my question wasn't clear, why don't you use/calculate MVD or MMD from the airborne measurements to compare with your measurements? If you have to stay with Deq, please also add the information to the text that it is used for comparability purposes as well.

p18, l12-32: Is there no reference for the IKP?

**I am afraid not.**

*I* am a little surprised because there are references for a probe named IKP2 (for example mentioned in one of your references – Leroy, 2017 - for the HAIC campaign, of which you also show data). Is this not related to the Cranfield IKP?

p23, l7: "...higher concentrations of small particles..." – could this not be due to particle shattering introducing numerous small particles (as the removal treatment was turned off)?

**CHRISTIANE & TINA. Please comment.**

Yes, particle shattering was taken care of by using anti-shattering tips and inter-particle time analysis ("removal treatment"). See also the above comment.

This is true for the field observations, but not for the wind tunnel!

In an earlier comment you said: "... Additionally the high concentrations in the wind tunnel would have rendered an inter-particle time analysis difficult because the inter-particle time of shattered and non-shattered particles were similiar...." This tells me, you did not use the inter-particle time analysis to identify possible shattering on your wind tunnel data, please clarify! Were anti-shattering tips used in the wind tunnel, if yes, add this information to the text.

My main point/question here is: How can you compare measurement data that use shattering artefact removal with data that are not treated to remove shattering artefacts? Particularly if you discuss differences in the number concentrations of small particles, which could caused by shattering artefacts. You also make the point (in your reply to my comment whether the p14, l31 – p15, l14 could be shortened) **"One valuable aspect of the paper is that it tries to compare wind tunnel and atmospheric conditions using the same instrument and evaluation technique thus making it independent on technical or computational aspects."** I would argue that when you use shattering removal on the field observationn data but not use it for the wind tunnel data, then you are not using the same evaluation technique! I think the implications of this should be discussed.

**Figures:**

I am still not sure whether Figure 5 and 35 are necessary, what value do they add to the manuscript?

*Figure 14: Caption:* "... dependence on ambient air temperature."  $\rightarrow$  "... *in dependence of ambient*..."

Figures 20 and 21: They show both spatial distributions for mixed phase cloud conditions, that's why I thought and still think they could be combined.

*Figure 23, right: x-axis says* "MVD vs. V water in ml min –1", *captions says* "MVD vs. LWC"?

Figure 27: Where was the IKP mounted?

Figures 29 and 30: If these figures are only aimed at showing examples of particles taken from different probes at different conditions, then I belive they could be combined. In the text explaining figure 30 (p23, l6/7) you say: "Based on the analyses of ice particle images one can conclude that the particles partially break up during conveyance and are reduced to smaller sizes."

You mean during the distribution step of the ice particles into the tunnel, or break up during the measurement at the cloud probe tips?

---

## Author Response (AR3)

Dear Dr. Frey:

On behalf of my coauthors, I thank you for reviewing the revised manuscript. Your comments are very valuable for us in order to improve our manuscript. Please find our answers to your questions below, typeset in blue color.

Kind regards,

Dr. S. Bansmer
* * *
p1, l8: "... than usually used..." – what is usually used?
**We stayed with the original wording, which came from the reviewer (British native speaker). Is that okay with you?**
*I did not say that the wording needs changing, but it needs explanation! You are talking about facilities „usually used" but it is not clear what facilities are usually used, name it!*

"Usually used" was referring to ice particles. To avoid this misconception, we will rephrase as follows:

*The unique aspect of this facility is the combination of an icing tunnel with a cloud chamber system for making ice particles. These ice particles are more realistic in shape and density than those usually used for mixed phase and ice crystal icing experiments.*

The usually system up to now was to shave/crunch an ice block into tiny ice particles. Such systems do not create the irregular structure of ice particles, which are found in the atmosphere.

p1, l9: "... measurement techniques for particle sizing have performed..." Either "have been performed" or "are performed".
**We stayed with the original wording, which came from the reviewer (Canadian native speaker). The meaning is "The techniques have performed well." Is that okay with you?** *The reviewer's suggestion was „… for particle sizing are performed on ice crystals" (which was one of my alternatives, too). If that does not reflect your meaning of the sentence it might be better to rephrase the sentence!*

We chose "are performed".

p5, l11: "Figure 46" – figures should be numbered in the order they appear in the text!
**We have put the large image on the facility deliberately as the last picture in the manuscript. I guess it depends how the final paper is typesetted... That's why the figure numbering is a bit odd at this place. We can move the figure of course, if you insist...**
*As I said, figures should be numbered in the order they appear. If you want tthe figure to appear at an exposed position, you could, I assume, easily move it to number 1 by refering to it early in your manuscript.*

We have moved the figure position to be consistent with the text. It is now figure 4.

p7, l16: figure13 – again, figures should be numbered in the order they appear in the text!

**Here it is more difficult. The Figure 13 is explained properly much later in the manuscript (section 2.9). It was just one reviewer, who wanted to have information on where the heat exchanger is placed inside the tunnel's circuit. That's why we created the link here. What do you suggest?**

*As I said, figures should be numbered in the order they appear. You could for example add a reference to the section where the figure is described in more detail here!*

*We added a reference to the section, where the figure is described.*

p12, l9: Is there a reference for TAUICE?

**No**

*Is it an abbreviation for something then? I feel there should be a few words about the code to give the reader an idea what it is used for etc.*

We added: "TAUICE is developed by the German Aerospace Centre. It solves the Reynolds-averaged Navier-Stokes equations using a finite volume approach, calculates droplet trajectories with a Lagrangian scheme and finally determines the ice accretion using a model of Messinger."

p14, l32/Figure 16: In the introduction you introduce Median Volume/Mass Diameters as important parameters, why do you use d eq here?

**The area equivalent diameter Deq defined as the diameter of a circle of the same area as the shaded pixelnumber for each particle is one way to deal with size information of particles. The information on particle shape is lost for Deq but hasn't been the focus of this study. In general, comparibility to the airborne measurements was aimed at and therefore Deq has been used. We added this information in the manuscript.**

*Maybe my question wasn't clear, why don't you use/calculate MVD or MMD from the airborne measurements to compare with your measurements? If you have to stay with Deq, please also add the information to the text that it is used for comparability purposes as well.*

In the field of engineering icing research, usually simple scalars like MVD and MMD are used to quantify the size information of a particle cloud. That is why we used MVD and MMD in the list of boundary conditions in the introduction.

Sometimes, it is however very useful to study how a particle cloud is distributed in terms of particle sizes in order to show its composition. Figure 16 (now Figure 14) demonstrates that the number concentration increases with increasing temperature for large particles, however, for small particles it is the opposite effect. An MMD could not reveal that information.

Our MVD and MMD are based or rather determined with the Deq. This is an important relation because the use of different diameter definitions changes the MVD and MMD. MMDs of airborne and wind tunnel tests have been compared in figure 29 (new manuscript). Since the MMDs of airborne and wind tunnel measurements do not agree in our study, the differences can only be clarified by comparison of size distributions. MMDs or MVDs for example do not give information on whether the size distribution is unimodal or bimodal. This means MMDs of the same magnitude could evolve from different size distributions. Moreover, we already had implemented the information you requested in figure 29 (new manuscript): MMD of airborne measurements vs. MMD of the icing wind tunnel setup.

We added the sentence: *With the distribution of $d_{eq}$ it is therefore possible to compare the composition of icing clouds in both atmosphere and icing wind tunnel environment.*

p18, l12-32: Is there no reference for the IKP?

**I am afraid not.**

*I am a little surprised because there are references for a probe named IKP2 (for example mentioned in one of your references – Leroy, 2017 - for the HAIC campaign, of which you also show data). Is this not related to the Cranfield IKP?*

*No, the IKP that was used in the atmospheric measurements (Leroy 2017) was developed by the Canadian NRC. There are indeed several publications by NRC on their IKP. However, Cranfield has not previously published an article on its IKP.*

p23, l7: "...higher concentrations of small particles..." – could this not be due to particle shattering introducing numerous small particles (as the removal treatment was turned off)?

**Yes, particle shattering was taken care of by using anti-shattering tips and inter-particle time analysis ("removal treatment"). See also the above comment.**

*This is true for the field observations, but not for the wind tunnel!*

*In an earlier comment you said:* „**... Additionally the high concentrations in the wind tunnel would have rendered an inter-particle time analysis difficult because the inter-particle time of shattered and non-shattered particles were similiar….**" *This tells me, you did not use the interparticle time analysis to identify possible shattering on your wind tunnel data, please clarify! Were anti-shattering tips used in the wind tunnel, if yes, add this information to the text.*

*My main point/question here is: How can you compare measurement data that use shattering artefact removal with data that are not treated to remove shattering artefacts? Particularly if you discuss differences in the number concentrations of small particles, which could caused by shattering artefacts. You also make the point (in your reply to my comment whether the p14, l31 – p15, l14 could be shortened)* „**One valuable aspect of the paper is that it tries to compare wind tunnel and atmospheric conditions using the same instrument and evaluation technique thus making it independent on technical or computational aspects.**" *I would argue that when you use shattering removal on the field observationn data but not use it for the wind tunnel data, then you are not using the same evaluation technique! I think the implications of this should be discussed.*

We added to the text: *" The 2D-S probe used during the airborne and wind tunnel measurements was equipped with anti-shattering tips to reduce shattering of large particles on the leading edge of the probe arms."*

We would like to demonstrate the effect of ice particle size on shattering with Figure 8a from Field et al., 2006:

[Figure]

Shown is the fraction of short interarrival times over the characteristic size (third moment of the size distribution devided by the second moment). A relevant fraction (> 10%) only appears for characteritic particle sizes larger that 1 mm.

These are data with a 2DC probe without anti- shattering tips. Thus the effects on the fraction of short interarrival times (shattered particles) should be an upper limit for the fraction of particles shattered on the 2DS arm equipped with anti-shattering tips. As shown in the PSDs in Figure 29 and the images in Figure 28, hardly any large particles were present in the icing wind tunnel (very few particles were larger than the maximum array width of 1.28 mm). We stress therefore that shattering of ice particles in the wind tunnel with a low number of large particles resulting in low characteristic particle sizes has only a minor effect on shattering and therefore on the measurements.

We added the sentence:
*"While most shattering should be minimized by the anti-shattering tips, the shattering treatment in regions of large effective diameters and high IWC further corrects for shattering artefacts (Korolev et al. 2013).* **Shattering is a function of IWC and effective diameter, see Field et al., (2006).** *Since hardly any particles were larger than the full array width (Figure 28), we argue that large particles that would have caused shattering artefacts are not present in the icing wind tunnel. Therefore the shattering treatment was turned off. "*

We remove the sentence in order not to cause too much confusion:
*"Additionally, the high concentrations in the wind tunnel would have rendered an inter-particle time analysis difficult because the inter-particle time of shattered and non-shattered particles were similar."*

*Figures:*
*I am still not sure whether Figure 5 and 35 are necessary, what value do they add to the manuscript?*

We decided to remove these figures.

*Figure 14: Caption: „… dependence on ambient air temperature." → „… in dependence of ambient…"*

We have changed the wording.

*Figures 20 and 21: They show both spatial distributions for mixed phase cloud conditions, that's why I thought and still think they could be combined.*

We have combined them and slightly changed their reference in the text.

*Figure 23, right: x-axis says „MVD vs.  water in ml min −1"VV   , captions says „MVD vs. LWC"?*

We changed the caption to "MVD vs. volumetric flow rate of liquid water at the spray system."

*Figure 27: Where was the IKP mounted?*

The IKP measurements were part of another measurement campaign, which was before the cross comparison of various optical array probes. We used the IKP measurements to calibrate the icing wind tunnel – to have a reference. In consequence, the IKP is not added to the sketch in that figure.

In the text we have currently the sentence *"Based on the IKP calibration, a cross comparison of various optical array probes has been performed in the Braunschweig Icing Wind Tunnel."*, which reflects the above statement.

The IKP itself was mounted at the tunnel centerline. We added that information in the text.

*Figures 29 and 30: If these figures are only aimed at showing examples of particles taken from different probes at different conditions, then I belive they could be combined. In the text explaining figure 30 (p23, l6/7) you say: „Based on the analyses of ice particle images one can conclude that the particles partially break up during conveyance and are reduced to smaller sizes."*
*You mean during the distribution step of the ice particles into the tunnel, or break up during the measurement at the cloud probe tips?*

We have combined the two figures. With the particle breakup process, we are referring to the conveyance system and not to breakup process at the cloud probe tips. The probe tips have been designed to avoid particle shattering (at least of those particles entering the measurement volume).

[revised manuscript text omitted]

md = 24.8 µm   md = 87 µm   md = 149 µm   md = 350 µm

$T_\infty = 0\,°\mathrm{C}$

md = 40.7 µm   md = 118 µm   md = 204 µm   md = 284 µm

$T_\infty = -5\,°\mathrm{C}$

[Figure]

1.28 mm

**Figure 28.** Ice particle images captured  at the  test section of the Braunschweig Icing Wind Tunnel. *Top:* Images taken by C-HSI probe. *Bottom:* Images taken by 2D-S probe, MMD = 79 µm, IWC = 3.2 g m$^{-3}$, $T_\infty = -15\,°\mathrm{C}$.

[Figure]

**Figure 29.** Comparison of artificial ice particle cloud to atmospheric cloud conditions. *Top:* Particle size distribution. *Bottom:* Cumulative particle size distribution.

[Figure]

**Figure 30.** Experimentally observed SLD ice shapes on a NACA0012 leading edge in the parameter space of $f_0$ and $A_c$, Re=$2 \cdot 10^6$, MVD=80 μm, AoA=0°. Results reproduced from Steiner and Bansmer (2016).

[Figure]

$T_\infty = 0\,°C$

$T_\infty = -5\,°C$

[Figure]

$T_\infty = -15\,°C$

**Figure 31.** Side view of ice accretion shapes on a cylinder for TWC=$12\,\mathrm{g\,m^{-3}}$, $m_r = 0.12$ and $U_\infty = 40\,\mathrm{m\,s^{-1}}$, with a variation of temperature.

[Figure]

[Figure]

[Figure]

**Figure 32.** Side view of ice accretion when heating the cylinder, $T_\infty = -5\,°C$, TWC=$6.8\,\mathrm{g\,m^{-3}}$ and $m_r = 0.5$.

---

## Author Response (AR4)

Dear Dr. Frey:

Regarding your comment about the shattering treatment, I have spoken with the responsible coauthor. Please find the answer below, typeset in blue colour.

We further found a small error in the scaling of the microscopic images in figure 15, which is corrected in the latest manuscript version.

We thank you again for your support during the review phase of our manuscript.

Kind regards,

Dr. S. Bansmer

I only have one point left where I would suggest to keep the sentence you removed now from the manuscript (starting "The particle sizes in the wind tunnel..."):

p20, l2-6:

"... Shattering of large particles is a function of IWC, see Field et al. (2006). The particle sizes in the wind tunnel were rather small (MMD around 80 µm) thus the fraction of shattered particles is expected to be less than 1%. Since furthermore hardly any particles were larger than the full array width, see the results in section 4.6.4, we argue that large particles that would have caused shattering artefacts are not present in the icing wind tunnel. Therefore the shattering treatment was turned off."

Maybe you could reduce potential confusion by rephrasing the second removed sentence similar to:

"Additionally, the high concentrations in the wind tunnel would have rendered an interparticle time analysis difficult because the inter-particle times of the wind tunnel particles are generally small and hard to distinguish from those of possible shattering artefacts."

That information could be kept before the sentence starting "Therefore, the shattering treatment...", I leave this up to you.

We still would like to remove the sentence. "Additionally, the high concentrations in the wind tunnel would have rendered an inter-particle time analysis difficult because the inter-particle time of shattered and non-shattered particles were similar."

We feel that this sentence might suggest that we wanted to avoid an inter-particle time analysis because it were a difficult task. However, a difficult task shall not be an excuse for not undertaking the necessary effort. Since we provided some good arguments in the subsequent sentence *"Since hardly any particles were larger than the full array width, we argue that large particles that would have caused shattering artefacts are not present in the icing wind tunnel."*, another reference to an inter-particle time analysis seems not necessary from our point of view.